

**Oligocene–Miocene paleoceanography off the Wilkes Land Margin**
**(East Antarctica) based on organic-walled dinoflagellate cysts**
Peter K. Bijl[1]*, Alexander J. P. Houben[2], Julian D. Hartman[1], Jörg Pross[3], Ariadna
Salabarnada[4], Carlota Escutia[4], Francesca Sangiorgi[1]
1 Marine Palynology and Paleoceanography, Laboratory of Palaeobotany and
Palynology, Department of Earth Sciences, Faculty of Geosciences, Utrecht University.
P.O. Box 80.115, 3508 TC Utrecht, The Netherlands
2 Applied Geoscience Team, Netherlands Organisation for Applied Scientific Research
(TNO), Princetonlaan 6, 3584 CB, Utrecht, The Netherlands
3 Paleoenvironmental Dynamics Group, Institute of Earth Sciences, University of
Heidelberg, Im Neuenheimer Feld 234, 69120 Heidelberg, Germany
4 Instituto Andaluz de Ciencias de la Tierra, CSIC-UGR, 18100 Armilla, Spain
* to whom correspondence should be addressed.
Email: p.k.bijl@uu.nl; phone +31 30 253 9318





**Abstract**
**Next to atmospheric $CO_2$ concentrations, oceanographic conditions are a critical**
**factor determining the stability of Antarctic marine-terminating ice sheets. The**
**Oligocene and Miocene epochs (~34–5 Ma) were time intervals with**
**atmospheric $CO_2$ concentrations between those of present-day and those**
**expected for the near future. As such, these time intervals may bear**
**information to resolve the uncertainties that still exist in the projection of**
**future ice-sheet volume decline. We present organic-walled dinoflagellate cyst**
**(dinocyst) assemblages from chronostratigraphically well-constrained**
**Oligocene to mid-Miocene sediments from Integrated Ocean Drilling Program**
**Expedition (IODP) Site U1356. Situated offshore the Wilkes Land continental**
**margin, East Antarctica, the sediment core has archived past dynamics of an ice**
**sheet that is today mostly grounded below sea level. We interpret dinocyst**
**assemblages in terms of paleoceanographic change on different time scales, i.e.,**
**on glacial-interglacial and long-term variability. Sea-ice indicators occur only**
**for the first 1.5 Ma following the full Antarctic continental glaciation during the**
**early Oligocene, and after the Middle Miocene Climatic Optimum. During the**
**remainder of the Oligocene and Miocene dinocysts suggest a weaker-than-**
**modern sea-ice season. The assemblages generally bear strong similarity to**
**present-day open-ocean, high-nutrient settings north of the sea ice edge, with**
**episodic dominance of temperate species similar to the present-day subtropical**
**front. Oligotrophic and temperate surface waters prevailed over the site**
**notably during interglacial time intervals, suggesting that the position of the**



**(subpolar) oceanic frontal systems have varied in concordance with Oligocene-**
**Miocene glacial-interglacial climate variability.**

**1. Introduction**
The proportion of the East Antarctic ice sheet that is presently grounded
below sea level is much larger than originally assumed (Fretwell et al., 2013). This
implies that much more ice is sensitive to basal melting by warm waters than
previously thought (Shepherd et al., 2012; Rignot et al., 2013; Wouters et al., 2015),
and that a much higher amplitude and faster rate of sea-level rise under future
climate scenarios than previously thought (IPCC, 2013). Studying the state and
variability of Antarctic ice volume during past episodes with high atmospheric $CO_2$
concentrations ($p CO_2$) might provide additional understanding into ice/ocean
feedback processes. Foster and Rohling (2013) compared sea-level and atmospheric
$p CO_2$ concentrations on geological timescales and highlighted that global ice sheets
were rather insensitive to climate change under atmospheric $p CO_2$ between 400 and
650 parts per million in volume (ppmv). During the Oligocene and Miocene
atmospheric $p CO_2$ ranged between 400 and 650 ppmv (Foster et al., 2012; Badger et
al., 2013; Greenop et al., 2014). Crucially, similar $p CO_2$ levels are expected for the near
future given unabated carbon emissions (IPCC, 2013), implying that global ice volume
may not change much under these $p CO_2$ scenarios.
In contrast to the invariant global ice volume inferred by Foster and Rohling
(2013), a strong (up to 1 per mille; ‰) variability is observed in deep-sea benthic
foraminiferal oxygen isotope (hereafter benthic $\delta^{18}O$) data (Pälike et al., 2006;
Beddow et al., 2016; Holbourn et al., 2007; Liebrand et al., 2011; 2017). These



benthic $\delta^{18}O$ data reflect changes in continental ice volume (notably on Antarctica), in
combination with deep-sea temperature, with the latter strongly coupled to polar
surface-water temperature, as deep-water formation was predominantly located at
high latitudes (Herold et al., 2011). High-amplitude variations in benthic $\delta^{18}O$ thus
suggest either (I) strong climate dynamics in the high latitudes with relatively minor
ice-volume change (which is in accordance with numerical modelling experiments
(Barker et al., 1999) and the inferences of Foster and Rohling (2013)), or (II) strong
fluctuations of the Antarctic ice-volume, with relatively subdued temperature
variability (which is in accordance with indications for an unstable Antarctic ice
sheets under warmer-than-present climates (Cook et al., 2013; Greenop et al., 2014;
Rovere et al., 2014). Indeed, if one assumes present-day $\delta$-composition (-42‰ versus
standard mean ocean water (SMOW)) for the Oligocene–Miocene Antarctic ice-sheets
and modern deep water temperature (2.5°C), then the Oligocene–Miocene benthic
$\delta^{18}O$ fluctuations suggest long-term ice-sheet-variability ranging between a present-
day size for 27–23 Ma and absence during numerous other time intervals (Liebrand
et al., 2017). Meanwhile, deep-sea temperatures have fluctuated considerably on
geologic time scales (as is evident from ice-free geologic episodes –e.g., Zachos et al.,
2008), suggesting there is no reason to assume that it did not fluctuate during the
Oligocene or Miocene as well. Therefore, likely a combination of deep-sea
temperature and ice-volume changes is represented in these records, but it is
intrinsically impossible to determine the relative contribution of both factors from
benthic $\delta^{18}O$ data alone. Clearly, ice-proximal reconstructions of climate, ice sheet and
oceanographic conditions are required to provide an independent assessment of the
stability of ice sheets under these $p$CO$_2$ conditions.



While the Oligocene–Miocene may, in terms of $pCO_2$ conditions, bear analogy
to our future, any such investigation must take into account the uncertainties
involved in Antarctic paleotopography, which determines the proportion of marine-
based versus land-based ice during the Oligocene. A lower Antarctic continent would
result in more ice sheets being potentially sensitive to basal melt, and as such a
higher sensitivity of the ice sheet to climate change. On top of this, one should take
note of the fundamentally different paleogeographic configuration of the Southern
Ocean during that time as compared to today (Figure 1). The development and
strength of the Antarctic Circumpolar Current (ACC) connecting the Atlantic, Indian
and Pacific Ocean basins (Barker and Thomas, 2004; Olbers et al., 2004) depend on
the basin configuration (width and depth of the gateways and position of continental
landmasses). The exact timing when the ACC reached its modern-day strength is still
uncertain, ranging from the Middle Eocene (41 Ma) to as young as Miocene (23 Ma,
Scher and Martin, 2004; Hill et al., 2013; Scher et al., 2015). Whether, and if so, how
the development of the ACC has influenced latitudinal heat transport, ice-ocean
interactions and the stability of Antarctic continental ice remains even more elusive.
To directly assess the role of ice-proximal oceanography on ice-sheet stability
during the Oligocene–Miocene, ice-proximal proxy-records are required. Several
ocean drilling efforts in the past have been undertaken to provide insight in the
history of the Antarctic ice sheets (Cooper and O'Brien, 2004; Barker et al., 1998;
Wise and Schlich, 1992; Barrett, 1989; Robert et al., 1998; Wilson et al., 2000;
Harwood et al., 2006; Exon et al., 2004; Escutia et al., 2011a). For some of these
sedimentary archives, establishment of age control was particularly challenging due
to the paucity of useful and proper means to calibrate the record to the international



time scale. As a consequence, their full use for the generation of paleoceanographic
proxy records and ice sheet reconstructions has remained limited.

In 2010, Integrated Ocean Drilling Program (IODP) Expedition 318 drilled an

inshore-to-offshore transect off Wilkes Land (Fig. 1a), a sector of East Antarctica that
is assumed to be highly sensitive to continental ice-sheet melt (Escutia et al., 2011b).
The sediments recovered from IODP Hole U1356A are from the continental rise of
this margin (Escutia et al., 2011b) and hence contain a mixture of shelf-derived
material and pelagic sedimentation. Dinoflagellate cyst events in this record have
been accurately tied to the international time scale through integration with
calcareous nannofossil, diatom and magnetostratigraphic data (Bijl et al., in press).
The result is a – for Southern Ocean standards – solid stratigraphic age frame for the
Oligocene–Miocene part of the record of Hole U1356 (Fig. 2; Table 1). In this paper,
we investigate the dinocyst assemblages from this succession by utilizing the strong
relationships between dinocyst assemblage composition and surface-water features
of today's Southern Ocean (Prebble et al., 2013). We reconstruct the oceanographic
regimes during the Oligocene and mid-Miocene, and speculate on their implications
for oceanographic settings. We further compare the palynological data with detailed
sedimentological descriptions from Salabarnada et al. (submitted this volume).
Pairing the sedimentological interpretation and biomarker-derived absolute sea
surface temperature (SST) reconstructions from the same core (Hartman et al.,
submitted this volume) with our dinocyst assemblage data, we assess the
oceanographic variability off Wilkes Land from the dinocyst assemblages both at
glacial-interglacial and long-term times scales.



**2. Material**

2.1 Site description for IODP Hole U1356A

Samples were taken from IODP Hole U1356A, drilled on the continental rise of

the Wilkes Land Margin, East Antarctica (Figure 1a; present coordinates 63°18.6' S,
135°59.9' E; Escutia et al., 2011b). We use the paleolatitude calculator
www.paleolatitude.org of van Hinsbergen et al. (2015) to reconstruct the
paleolatitudinal history of the site (Figure 1, between -59.8±4.8°S and -61.5±3.3° S
between 34 Ma and 13 Ma, respectively). The single hole at Site U1356 reaches a
depth of 1006.4 m into the seabed (Escutia et al., 2011b). Oligocene to late Miocene
sediments were recovered between 890 and 3 mbsf (Figure 2; Tauxe et al., 2012;
revised according to Bijl et al., in press). The uppermost 95 meters of the hole were
poorly recovered; sediments consisted of unconsolidated mud strongly disturbed by
rotary drilling (Escutia et al., 2011b). Hence, we focused our investigation on the
interval between Cores 11R to 95R Section 3 (95.4 to 894 mbsf; 10.8-33.6 Ma; Figure

2).


2.2 Lithology in IODP Hole U1356A

In the studied interval between 95.4 and 894 mbsf, nine lithologic units have

been recognized during shipboard analysis (Figure 2; et al., 2011b). Salabarnada et al.
(submitted this volume) presents a detailed lithologic study of the Oligocene
sediments. For the grouping of our results, we use the lithologic facies from
Salabarnada et al. (submitted this volume), as outlined in Table 2. For the Miocene
interval of Site U1356, such a detailed lithologic description is not yet available;
therefore we treat the Miocene sediments as one separate lithologic unit in this



paper. For the Miocene, we here give a brief summary of the observations published
in the IODP Expedition 318 post-cruise report (Escutia et al., 2011b). Miocene
sediments between 95 and 400 mbsf reflect increasing consolidation down-core, and
comprise diatom ooze and diatom-rich silty clays. The more consolidated bedding has
caused better preservation of original bedding structures. From 278.4 to 459.4 mbsf,
the lithology lacks gravel-sized clasts, but is otherwise similar to up-core.

2.3 Bio-magnetostratigraphic age model for IODP Hole U1356A

Stratigraphic constraints for the Oligocene–Miocene succession from IODP

Hole U1356A are provided through calcareous nannoplankton, radiolarian, diatom
and sparse palynological biostratigraphy, complemented with magnetostratigraphy
(Tauxe et al., 2012). Bijl et al. (in press) and Crampton et al. (2016) have updated the
existing age model for Site U1356 for the Oligocene and Miocene part of the
succession, respectively. Thereby, they recalibrated to the international time scale of
Gradstein et al., 2012. We here follow these new insights of the age model (Table 1).
We infer ages by linear interpolation between tie points (Figure 2; Table 1).

2.4 Depositional setting IODP Site U1356

The depositional setting of Site U1356 changed from a shallow mid-

continental shelf in the early Eocene (Bijl et al., 2013a) to a deep continental rise
setting in the Oligocene (Houben et al., 2013) due to subsidence of the Wilkes Land
Margin (e.g., Close et al., 2009). Regional extrapolation of the lithology at U1356A via
seismic profiles suggests a mix of distal-fan and hemipelagic sedimentation during
the early Oligocene, grading into channel-levee deposits towards the later Oligocene



(Escutia et al., 2011b). The boundary between these two different depositional
settings occurs at ~650 mbsf; there, sedimentation rates increase, and the
documentation of mass-transport deposits from this depth upwards suggest shelf-
derived erosion events on the Wilkes Land continental slope (Escutia et al., 2011b).

**3. Methods**

3.1 Palynological sample processing

We refer to Bijl et al. (in press) for sample processing and analytical

procedures used. Both were according to standard procedures (e.g., Bijl et al., 2013b).
The 25 species of dinocysts new to science, which are formally (2 species) and
informally (23 species) described in Bijl et al. (in press) fit into known and extant
genera and therefore could be confidently included in the ecological groups as
described below.

3.2 Ecological grouping of dinocyst taxa

Bijl et al. (in press) provided additional statistical evidence to distinguish *in*

*situ* dinocysts from those that are reworked from older strata. In this paper, we follow
the interpretations of Bijl et al. (in press) and divide the dinocyst species into a
reworked and an *in situ* part (Table 3). To use the *in situ* dinocyst assemblages for
oceanographic reconstructions, we rely on the observation that many taxa in the
fossil assemblages have morphologically closely related modern counterparts. This
approach takes advantage of studies on present-day relationships between Southern
Ocean microplankton in general and dinoflagellates in particular and their surface-
water characteristics (e.g., Eynaud et al., 1999; Esper and Zonneveld, 2002, 2007;



Prebble et al., 2013). We assign Oligocene–Miocene dinocyst taxa to present-day eco-
groups interpreted from the clusters identified by Prebble et al. (2013), which seem
to be closely related to the oceanic frontal systems in the Southern Ocean (Figure 3).
Supporting evidence for the ecologic affinities for the dinocyst groups comes from
empirical data (Sluijs et al., 2005), for instance when it comes to the oceanic affinities
of *Nematosphaeropsis labyrinthus*, *Operculodinium* spp., *Pyxidinopsis* spp. and
*Impagidinium* spp. There is further abundant evidence, both empirically (e.g., Sluijs et
al., 2003; Houben et al., 2013) and from modern observations (Zonneveld et al., 2013;
Prebble et al., 2013; Eynaud et al., 1999), which link the abundance of
protoperidinioid dinocysts to high surface-water primary productivity. The arguably
most important inference from the surface-sample study of Prebble et al. (2013) is
that *Selenopemphix antarctica* is common to dominant (10-90%) in proximal sea-ice
settings south of the Antarctic polar front (AAPF). Notably, none of the surface
samples outside of the AAPF have dominant *Selenopemphix antarctica* (Prebble et al.,
2013). Another important observation is that the surface samples south of the AAPF
are devoid of gonyaulacean dinocysts, with the exception of two species of
*Impagidinium* (i.e., *I. pallidum* and *I. sphaericum*) which can occur, although neither
abundantly (Prebble et al., 2013) nor exclusively (e.g., Zevenboom, 1995; Zonneveld
et al., 2013), in ice-proximal locations. Another important observation is the
occurrence of abundant *Nematosphaeropsis labyrinthus* exclusively in regions outside
of the Subantarctic Front, and particularly close to the Subtropical Front. In summary,
from proximal Antarctic to outside the frontal systems, Prebble et al. (2013)
documents dominance of *S. antarctica* south of the AAPF, dominance of other
protoperidinioid dinocysts at and N of the AAPF, mixed protoperidinioid and





gonyaulacoid dinocysts (with a notable common occurrence of *Nematosphaeropsis*
*labyrinthus* at the SAF and mixed gonyaulacoid dinocysts at and outside of the STF.
These trends represents the transition from sea-ice influenced to cold upwelling/high
nutrient to warm-temperate/lower nutrient conditions, respectively. We use the
affinities obtained by Prebble et al. (2013) to reconstruct past oceanographic
conditions at the Wilkes Land continental margin.

**4. Results**

4.1 Palynological groups

In our palynological analysis we separated palynomorph groups into four

categories: *In situ* dinocysts, reworked dinocysts (following Bijl et al. (in press); Table
3), acritarchs and terrestrial palynomorphs. Our palynological slides further contain a
varying amount of pyritized diatoms and a minor component of amorphous
palynofacies, which is not further considered in this study. The relative abundance of
the four palynomorph groups varies considerably throughout the record, as do their
absolute abundances (Figure 4). Reworked dinocysts are present to common
throughout the record, but are particularly abundant in the lowermost 40 meters of
the Oligocene and in the Upper Oligocene. *In situ* dinocysts dominate the
palynomorph assemblage during the mid-Oligocene and mid-Miocene. Chorate,
spheromorph and *Cymatiosphaera*-type acritarchs (which are not further
taxonomically subdivided in this study) dominate the assemblage during the late
Oligocene and into the mid-Miocene, while terrestrial palynomorphs (which are
considered *in situ* and not reworked from older strata (Strother et al., 2017)) are a
constant minor component of the total palynomorph assemblage (Fig. 4).




4.2 *In situ* dinocyst assemblages
Throughout the Oligocene, *in situ* dinocyst assemblages are dominated by
protoperidinioid dinocysts, notably *Brigantedinium* spp., *Lejeunecysta* spp., *Malvinia*
*escutiana,* and *Selenopemphix* spp. (Figure 4), all of which are considered associated
with heterotrophic dinoflagellates. Among these protoperidinioid cysts, *S. antarctica*
is common to abundant only in the first 1.5 million years of the Oligocene
represented in the core material (33.6–32.1 Ma), and during and after the mid-
Miocene climatic transition (<14.2 Ma; Fig. 5). The remainder of the record is
generally devoid of *S. antarctica*. This is much in contrast to the dinocyst assemblages
at Site U1356 today, which are dominated by this taxon (Prebble et al., 2013). Instead,
other protoperidinioid dinocysts dominate, such as *Brigantedinium* spp., several
*Lejeunecysta* species and *Selenopemphix nephroides*, which have close affinities to
high-nutrient conditions in general (e.g., Harland et al., 1999; Zonneveld et al., 2013)
but are not specifically restricted to sea-ice-proximity or the Southern Ocean. Today,
these three genera dominate dinocyst assemblages in high-nutrient regions at or
outside of the AAPF (Prebble et al., 2013). We also encountered a varying abundance
of protoperidinioid dinocysts, which could not be placed with confidence into
established protoperidinioid dinocyst genera. These are grouped under
protoperidinioid spp. pars (Figure 4), and are here assumed to exhibit the same
heterotrophic life-style as the other protoperidinioid dinocyst genera.
Next to peridinioid dinocysts, also gonyaulacoid dinocysts occur commonly to
abundantly throughout the record from Site U1356. They comprise both known and
previously unknown (Bijl et al., in press) species of *Batiacashaera, Pyxidinopsis,*





*Nematosphaeropsis, Impagidinium,* and *Operculodinium* (Fig. 4; 5). Except for the
extinct genus *Batiacasphaera*, all the other genera are still extant and are formed by
phototrophic dinoflagellates. The abundance of these presumably mostly autotrophic
taxa (Zonneveld et al., 2013) goes at the expense of the assumed heterotrophic
protoperidinioid dinocysts. A remarkable increase is noted associated with the mid-
Miocene Climate Optimum (between ~17 and 15 Ma; Fig. 4, 5; Sangiorgi et al., in
review). Of these taxa, *Nematosphaeropsis* is thought to be associated with frontal
systems of the present-day Southern Ocean (Prebble et al., 2013) and also in the
North Atlantic Ocean  (Boessenkool et al., 2001; Zonneveld et al., 2013).

4.5 Comparison between palynological data and lithological interpretations
The Oligocene sediments from Site U1356 comprise distinctive alternations of
lithologic facies throughout the section (Salabarnada et al., submitted this volume;
Figure 2). They are interpreted to reflect changes in the oceanographic regime, with
relations to glacial-interglacial changes (Salabarnada et al., submitted this volume).
Carbonate deposits, pelagic claystones and bioturbated, carbonate-bearing silty
claystones were interpreted as interglacial deposits, while the laminated lithologies
reflect glacial deposits (Salabarnada et al., submitted this volume). Mass-transport
deposits reflect times of major sediment transport from the continental shelf. The
lower Oligocene glauconitic sandstones were interpreted to reflect episodes of
redeposition of winnowed upper Eocene shelf sediments (Sluijs et al., 2003; Houben,
2012). We here evaluate and compare the palynological content of each of these
lithologies, both in terms of absolute and relative abundance of the main



palynomorph groups: reworked dinocysts, *in situ* dinocysts, acritarchs and terrestrial
palynomorphs and relative abundance of *in situ* dinocyst eco-groups.

4.5.1 Palynomorph groups and lithology

There are distinct differences in the relative and absolute abundances of

palynomorph groups between the different lithologies (Figure 6). The highest relative
and absolute abundances of reworked dinocysts occur in the lower Oligocene
reworked glauconitic sandstones, which is in line with previous inferences of Houben
et al. (2013). The mass-transport deposits contain abundant reworked dinocysts. The
relative and absolute abundance of *in-situ* dinocysts does not vary much between the
different lithologies, with the exception of the pelagic clays, in which *in situ* dinocysts
are much lower in relative and absolute abundance (Figure 6). The opposite pattern
emerges for acritarchs, which reach highest relative and absolute abundances in the
pelagic clays (Figure 6). Terrestrial palynomorphs are most abundant in the
glauconitic contorted sandstones (Figure 6).

4.5.2 *In situ* dinocyst eco-groups and lithology

We also compared the *in situ* dinocyst eco-groups with predominant

lithological facies (Figure 7). The abundance of *Selenopemphix antarctica* is low
throughout the record (0-5%), with the exception of the interval post-dating the
Miocene Climatic Optimum (MCO) interval and the lowermost Oligocene. We note
that in the lower Oligocene, high abundances of *S. antarctica* and *Malvinia escutiana*
are mostly connected to glauconitic sandstones and the mass-transport deposits, and
rarely occur in the other lithologies (Figure 7). We however think that these species





represent part of the *in situ* assemblage in an otherwise dominantly reworked
dinocyst assemblage, because these were never found in Eocene sediment in the
region before. *Lejeunecysta* spp. shows significantly higher relative abundances in the
mass-transport and glacial deposits, and substantially lower abundance in the pelagic
clays, interglacial deposits and in the Miocene. *Brigantedinium* spp. shows invariable
relative abundances in the different lithologies, and the *Protoperidinium* spp. pars
group shows highest abundance in the pelagic clays (Figure 7). Overall, the relative
abundances of all (proto)peridinioid dinocysts in the *in situ* assemblage is highest in
the glacial deposits and pelagic clays, and substantially lower in interglacial deposits
and in the Miocene. Indeed, several gonyaulacoid dinocyst taxa (such as
*Nematosphaeropsis* spp., *Pyxidinopsis* cpx, *Operculodinium* spp., and *Impagidinium*
spp.) show higher relative abundances in interglacial than in glacial deposits. We thus
observe a marked difference in the relative abundances of gonyaulacoid dinocysts
over peridinioid dinocysts between glacial and interglacial deposits.

**5. Discussion**

5.1 Paleoceanographic interpretation of the dinocyst assemblages

5.1.1 Surface-ocean nutrient conditions

The dominance of heterotrophic dinoflagellate cysts in the Oligocene-Miocene

dinocyst assemblages indicate overall high nutrient levels in the surface waters. We
infer therefore that in general, surface-waters overlying Site U1356 experienced
upwelling associated to the AAPF during most of the Oligocene and Miocene.
However, and surprisingly, the occasionally abundant oligotrophic cyst taxa
encountered in our record suggest that at times, surface waters were much less



nutrient-rich, supporting an oligotrophic dinoflagellate assemblage. These dinocysts
are outer shelf to oceanic or outer neritic taxa (e.g., Sluijs et al., 2005; Zonneveld et al.,
2013; Prebble et al., 2013), which makes it unlikely they were reworked from the
continental shelf. Indeed, these taxa show low relative abundances in the mass-
transport deposits (Figure 6); hence, we interpret that these taxa are part of the *in*
*situ* pelagic assemblage and reflect warming of surface waters rather than them being
reworked. Although species within these genera have relatively long stratigraphic
ranges extending back into the Eocene, most of the species encountered at U1356
have never been found in Eocene continental shelf sediments in the region (e.g., Bijl et
al., 2011; 2013a, b; Brinkhuis et al., 2003a, b; Levy and Harwood, 2000; Wrenn and
Hart, 1988). This lends further support against them being reworked from Eocene
shelf material, in addition, the statistical approach also interprets these species to be
part of the *in situ* assemblage (Bijl et al., in press). Now that we have abundant
evidence that these autotrophic taxa are part of the *in situ* pelagic assemblage, we can
interpret these assemblages in terms of their paleoceanographic affinities. The
occasional abundance of oligotrophic taxa suggests nutrient levels must have been
low compared to the same region today. The absence of these taxa in modern surface
waters south of the AAPF is probably caused by a combination of factors: low sea
surface temperatures, isolation by strong eastward currents, but also the abundance
and seasonal concentration of nutrients, which make the Antarctic proximal surface
waters a very specialistic niche. Apparently, surface water conditions during the
Oligocene and Miocene were such that these oligotrophic species could at times
proliferate so close to the Antarctic margin.





5.1.2 sea-surface temperature
The average dinocyst assemblages in our record point to the Southern margin of New
Zealand and Tasmania as the best modern analogue (inferred from Prebble et al.,
2013; Figure 2). Those regions today feature a mix between protoperidinioid
dinocysts and gonyaulacoid dinocyst genera such as *Nematosphaeropsis*,
*Operculodinium* and *Impagidinium*. These assemblages occur at present in surface-
waters with mean annual temperatures of 8-17°C (Prebble et al., 2013). A bayesian
approach on the $TEX_{86}$ index values at U1356 (presented in Sangiorgi et al.,
submitted; Hartman et al., submitted this volume) indicates exactly the same region
as modern analogues for the $TEX_{86}$ index values found (Hartman et al., submitted this
volume) as for the dinocysts (Prebble et al., 2013); both approaches indicate the same
paleotemperature range for the Oligocene-Miocene at U1356. These two proxies thus
independently point to a temperate, much warmer paleoceanographic regime close to
Antarctica during the Oligocene and Miocene with the nearest modern analogue
being offshore Southern New Zealand and Tasmania. Supporting evidence for
temperate Oligocene-Miocene surface waters comes from the abundance of
nannofossils encountered in the same Oligocene-Miocene sediments (Escutia et al.,
2011b). Today, carbonate-producing plankton is not abundant in high-latitude
surface waters south of the AAPF (Eynaud et al., 1999). Moreover, the remains of the
few carbonate-producing organisms living at high latitudes rarely reach the ocean
floor because strong upwelling of relatively $CO_2$-rich, corrosive waters (e.g., Olbers et
al., 2004). Hence, the presence of carbonate-rich intervals during the Oligocene-
Miocene at Site U1356, along with the encountered oligotrophic, temperate dinocysts,
suggests fundamentally warmer surface-water conditions than at present.




### 5.1.3 Paleoceanography

The strong similarity of Oligocene–Miocene dinocyst assemblages at Site
U1356 to those today occurring much further north (e.g., around Tasmania and
Southern New Zealand (Prebble et al., 2013) suggests a fundamentally different
*modus operandi* of Southern Ocean oceanography. The strict latitudinal separation of
dinocyst assemblages throughout the Southern Ocean today (Prebble et al., 2013) is
likely due to the different water masses present across the oceanic fronts where
strong wind-driven divergence around 60° S (known as the Antarctic Divergence; e.g.,
Olbers et al., 2004), strong sea-ice season and/or the vigorous Antarctic Circumpolar
Current are in place. The strength and position of the AAPF during the Oligocene–
Miocene is not well understood. GCM experiments under Miocene boundary
conditions suggest that west and east wind drifts prevailed south and north of 60°S,
respectively (Herold et al., 2011). This position of the winds determines the average
position of the Antarctic Divergence at 60°S during the Oligocene and Miocene, like
today. This would mean that Site U1356 likely was directly overlain by the AAPF.
However, the significantly warmer, more oligotrophic character of the dinocyst
assemblages offshore Wilkes Land throughout the Oligocene–Miocene argues against
a close position to the AAPF. The position of the AAPF relative to the position of Site
U1356 strongly determines the likelihood of southward transport of low-latitude
waters towards the site. A southward position of the AAPF relative to Site U1356
would greatly enhance the possibility for southward migration of temperate water
masses towards the site. A northward position of the AAPF relative to the site, would
make such much more difficult. The presence of carbonate in these deep marine



sediments also suggests that upwelling of corrosive waters through the (proto-)
Antarctic Divergence was either much reduced or located elsewhere. Therefore, we
deduce that the occurrence of the oligotrophic, temperate dinocysts is evidence for a
southward position of the AAPF relative to the position of Site U1356.

The separate averaging of dinocyst assemblages for glacial and interglacial

deposits (Figure 7) allows us to reconstruct the glacial-interglacial surface
oceanographic changes throughout the Oligocene. This approach suggests that
substantial paleoceanographic dynamics were associated with Oligocene glacial-
interglacial cycles. Alongside the 2–3 °C SST variability during glacial-interglacial
cycles at this same site (Hartman et al., submitted this volume), dinocyst assemblages
contain more oligotrophic, temperate dinocysts during interglacial time intervals
compared to glacial intervals when more eutrophic, colder dinocysts proliferated.
This could be the result of a slight latitudinal movement of oceanic frontal systems
(notably the AAPF), as has been reconstructed for the Southern Ocean fronts during
the most recent glacial to interglacial transition (e.g., Kohfeld, et al. 2013). The
difference in dinocyst assemblages between glacial and interglacial deposits might be
explained by a south position of the AAPF during interglacials, allowing for temperate
oligotrophic surface waters to reach the Site, while during glacials the AAPF migrated
northward over Site U1356, causing cold, high-nutrient conditions.

5.2 Implications for Oligocene-Miocene ocean circulation

Only in the lowermost Oligocene and in strata representing the mid-Miocene

climatic transition and later (14.4 Ma and younger), the dinocyst assemblages bear
similarities to modern proximal-Antarctic assemblages (Prebble et al., 2013), with



high abundances of *Selenopemphix antarctica*. Even in those intervals, however, the
relative abundances of *S. antarctica* does not reach present-day values at the same
site. The absence of a strong shift towards modern-day-like assemblages in our
record can be interpreted to reflect a weaker-than-present ACC, in line with
numerical models (Herold et al., 2012; Hill et al., 2013). The ACC itself represents an
important barrier for latitudinal surface-water transport towards the Antarctic
margin, in addition to the Antarctic Divergence (Olbers et al., 2004). Our data suggest
an increase in the influence of oligotrophic dinocysts at the Antarctic margin during
the late Oligocene and during the MMCO, which argues against the installation of a
vigorous ACC at 30 Ma (Scher et al., 2015): No profound changes in surface
paleoceanography emerge from our dinoflagellate cyst data around 30 Ma, and there
is no major change in the benthic $\delta^{18}O$ (Figure 5). Instead, if the Tasmanian Gateway
had opened to an extent that allowed ACC development (Scher et al., 2015), the ACC
must have been much weaker than at present throughout the Oligocene and Miocene.
The strongly different dinocyst assemblages compared to present-day at Site U1356
throughout our record implies to us that a strong coherent ACC was not installed until
after the MMCT (11 Ma). This is consistent with inferences from the lithology at the
same site (Salabarnada et al., submitted this volume), suggesting a proto-ACC much
weaker than at present and, likewise, weaker Southern Ocean frontal systems. An
alternative explanation is that the ACC increased in strength during the Oligocene–
Miocene, but that this strengthening had no influence on the dinocyst assemblages at
Site U1356. However, the vigorous nature of the ACC influencing surface as well as
bottom waters and governing eddy water circulation in the Southern Ocean (Olbers et
al., 2004) makes such a scenario very unlikely. Nevertheless, to firmly clarify whether



the strength of the ACC changed to its present-day force only after the MMCT (as
suggested by our data), ocean-circulation modelling of time slices younger than the
Oligocene will be required.

5.3 Implications for ice sheet and sea-ice variability

The abundance of our sea-ice indicator *Selenopemphix antarctica* throughout

the record is consistently lower than that in present-day dinocyst assemblages at Site
U1356 (Prebble et al., 2013; Figure 3). This suggests that sea-ice conditions were
never as severe as today throughout the studied time interval. Only during two time
intervals sea ice indicators suggest some sea ice near the Site: the first 1.5 million
years following the Oi-1 glaciation (33.6–32.1 Ma; Figure 5), and during and after the
mid-Miocene climatic Transition (14–11 Ma; Figure 5). Numerical ice-sheet/sea-ice
modelling (DeConto et al., 2007) suggests sea-ice to develop only if the continental ice
sheets reach the coastline. Our lack of sea-ice indicators during most of the Oligocene
and Miocene could thus suggest that the Antarctic continental ice sheet was much
reduced during this time. The finding of a weaker sea-ice season throughout most of
the Oligocene–Miocene at Site U1356 has major implications for regional
paleoceanography because it suggests a decrease in the potential formation of
Antarctic bottom waters at this site.

The abundance of our oligotrophic taxa broadly co-varied with long-term

Oligocene-Miocene benthic $\delta^{18}O$: During times of low $\delta^{18}O$ values in deep-sea benthic
foraminifera (and thus high deep-sea temperatures and less ice volume; e.g., at 32 Ma,
24 Ma and 15 Ma; Figure 5), the abundance of oligotrophic temperate dinocysts was
large (Figure 5). At times of higher $\delta^{18}O$ values, lower deep-sea temperatures and



higher ice volume (e.g. at 33.5 Ma, 27 Ma, 23 Ma and 13 Ma; Figure 5) temperate
dinocysts were reduced in abundance and high-nutrient, sea-ice indicators
(re)appeared. Altogether, this suggests on long time scales, that there was stronger
influence of warm surface waters at the Wilkes Land Margin at times when ice sheets
were smaller and climate was warmer, and less influence of warm surface waters
during times of larger ice sheets, hence a connection between ice sheet and
oceanographic variability.

Oxygen-isotope mass-balance calculations suggest that a modern-day-sized

Antarctic ice sheet appeared at the Eocene/Oligocene boundary (DeConto et al.,
2008). Benthic $\delta^{18}O$ records suggest that ice sheets fluctuated considerably in size
during the subsequent Oligocene and Miocene (Liebrand et al., 2017). Based on the
heavy $\delta^{18}O$ values for Oligocene benthic foraminifera from Maud Rise, it was inferred
that Antarctic ice sheets were near-present-day size throughout the Oligocene
(Hauptvogel et al., 2017). Both isotope studies of Liebrand et al (2017) and
Hauptvogel et al. (2017) assume constant temperatures of the deep sea and similar-
to-present-day $\delta^{18}O$ of the continental ice. Our data instead show that the regional
paleoceanography, together with surface-ocean temperature (Hartman et al.,
submitted this volume), can vary considerably both on the long term as on orbital
time scales. It remains to be seen whether the variability in paleoceanography found
here can be extrapolated to larger parts of the Antarctic margin, including to those
regions of deep-water formation. Given the high temperatures and absence of strong
sea ice influence, the Wilkes Land margin was likely not the primary sector of deep-
water formation, although there is ample evidence for bottom-current activity at the
site (Salabarnada et al., submitted this volume). However, if the oceanographic and



climate variability we reconstruct offshore Wilkes Land characterises also regions of
deep-water formation, some (if not much) of the variability both on long and on
orbital time scales in benthic $\delta^{18}$O records is related to deep-sea temperature rather
than Antarctic ice volume (see also Hartman et al., submitted this volume).
Meanwhile, we find little support in our study for the large continental ice sheets
during the Oligocene as concluded by Hauptvogel et al. (2017), given the absence of
dominance of sea-ice dinoflagellate cysts and *in situ* terrestrial palynomorphs
(Strother et al., 2017). As an alternative explanation to the difference in $\delta^{18}$O values
between Maud Rise and Equatorial Pacific during the Oligocene (Hauptvogel et al.,
2017), we suggest that these two records have recorded the characteristics of two
fundamentally different deep water masses, with those at Maud Rise being much
colder and saltier than those at Shatsky Rise.

**6. Conclusions**
The dinocyst assemblage changes in the Oligocene–Miocene (33.6–10 Ma) of Site
U1356 were interpreted in terms of surface paleoceanography based on a
comparison of these assemblages to present-day dinocyst assemblages. This
approach allows us to hypothesize that the Southern Ocean paleoceanography during
the Oligocene–Miocene was fundamentally different from that of today. A strong sea-
ice signal (yet still weaker than that of today) emerges for the Wilkes Land Margin
only for the first 1.5 million years of the Oligocene (33.6–32.1 Ma) and the mid-
Miocene climatic transition (14-10 Ma). The remainder of the Oligocene–Miocene
record of surface waters off Wilkes Land were warm, relatively oligotrophic and lack
indications of a prominent sea-ice season. Upwelling at the Antarctic Divergence must



have been profoundly weaker during Oligocene and Miocene times, compared to
today. Furthermore, the continental ice sheet must have been much reduced at the
Wilkes Land sub-glacial basin for most of the Oligocene-Miocene compared to today,
and continental ice sheets were retreated inland. The strength of the influence of
warm oligotrophic surface water was strongly coupled to deep-sea $\delta^{18}O$ values: With
enhanced low-latitude influence of surface water during times of light $\delta^{18}O$ in the
deep sea and *vice versa*. The absence of (a trend towards more) oceanographic
isolation of the Wilkes Land margin throughout the Oligocene to mid-Miocene
suggests that the ACC did not obtain its full, present-day strength until at least the
mid-Miocene Climatic transition. Moreover, we note considerable glacial-interglacial
variability in this oceanographic setting, with stronger influence of oligotrophic, low-
latitude surface waters over Site U1356 during interglacial times and more eutrophic,
colder influence during glacial times. This may suggest considerable latitudinal
migration of the AAPF over Oligocene and Miocene glacial-interglacial cycles.

**Acknowledgements**
This research used data and samples from the Integrated Ocean Drilling Program
(IODP). IODP was sponsored by the U.S. National Science Foundation and
participating countries under management of Joined Oceanographic Institutions Inc.
PKB and FS thank NWO-NNPP grant no 866.10.110, NWO-ALW VENI grant no
863.13.002 for funding and Natasja Welters for technical support. CE and AS thank
the Spanish Ministerio de Economía y Competitividad for Grant CTM2014-60451-C2-
1-P.



**Author contributions**

PKB, FS, CE and JP designed the research. AJPH, FS and PKB carried out dinoflagellate cyst analyses for the earliest Oligocene, the middle Miocene, and the Oligocene-Miocene boundary interval, respectively. AS and CE provided the lithological data. PKB integrated, cross-validated and compiled the data, and wrote the paper with input from all co-authors.



**Figure captions**
Figure 1 Paleogeography of the Southwest Pacific Ocean and position of IODP Site
U1356 (Red star) at (a) 0 Ma, (b) 10 Ma, (c) 20 Ma, and (d) 30 Ma. Figures were
modified from Bijl et al. (in press). Reconstructions were adapted from G-plates, with
plate circuit from Seton et al. (2012) and absolute plate positions of Torsvik et al.

585 (2012).


Figure 2. Age model for the Oligocene–Miocene interval of Hole U1356A. Core
recovery, lithostratigraphic units and log, age-depth plot (from Tauxe et al., 2012, but
recalibrated to GTS2012 of Gradstein et al., 2012; see Table 1 and modified based on
Crampton et al., 2016), and samples taken for palynology. Figure modified from Bijl et
al. (in press).

Figure 3. Generic representation of present-day distributions of dinocysts in surface
sediments in the Southern Ocean. The dinocyst pie charts represent average
dinoflagellate cyst assemblage compositions for surface sediments underneath
oceanic frontal zones in the Southern Ocean. Figure modified from Sangiorgi et al. (in
review), data replotted from Prebble et al. (2013).

Figure 4. Core recovery, lithostratigraphic log (after Salabarnada et al., this volume),
chronostratigraphic epochs (E = Eocene) and stages (L = Lutetian, Burd. =
Burdigalian, Ser. = Serravallian, T. = Tortonian), absolute palynomorph (grey) and *in*
*situ* dinocyst (black) concentrations (# per gram of dry sediment, presented on a
logarithmic scale), palynomorph content (reworked dinocysts, *in situ* dinocysts,



acritarchs, and terrestrial palynomorphs; given in percentages of total
palynomorphs), and relative abundance of *in situ* dinocyst assemblages (in
percentage of *in situ* dinocysts) for the Oligocene–Miocene of Hole U1356A.

Figure 5. Benthic foraminiferal oxygen isotope data from Site 588 (Zachos et al.,
2008), Site 1090 (Zachos et al., 2008) Site 1218 (recalibrated from (Pälike et al.,
2006), Site U1334 (Holbourn et al., 2015), Site U1337 (Beddow et al., 2016), with
Dinocyst assemblage data from Site U1356. We used the paleomagnetic tie points of
Tauxe et al. (2012) (with the exception of the Oligocene–Miocene boundary interval,
see text) recalibrated to Gradstein et al. (2012) for calibrating our data to age,
following the age-depth model specified in Figure 2 and Table 1.

Figure 6. Comparison of relative (left bar; in % of total palynomorphs) and absolute
(right bar, in # * gr $^{-1}$ dry weight) abundances of palynomorph groups per lithology.
Average (black lines) and standard deviation (coloured bars) of absolute and relative
abundances of total palynomorphs, reworked dinocysts, *in situ* dinocysts, acritarchs
and terrestrial palynomorphs grouped in the different lithologies: Miocene
sediments, carbonate deposits, bioturbated sediments, pelagic clays, laminated silty
claystones, laminated sand stones, mass-transport deposits and glauconitic sand
stones.

Figure 7. Comparison of *in situ* eco-groups with lithology. Average (black line) and
standard deviation (coulored bar) of relative abundances of grouped taxa from
samples from the different lithologies: Miocene sediments, carbonate deposits,



bioturbated sediments, pelagic clays, laminated silty claystones, laminated sand
stones, mass-transport deposits and glauconitic sand stones.

**Table captions**
Table 1. Age constraints for the Oligocene–Miocene of Hole U1356A.
Table 2. Lithologic facies described in Salabarnada et al. (submitted this volume), and
in this paper.
Table 3. List of assumed *in situ* and reworked dinoflagellate cyst taxa encountered in
this study. See Bijl et al. (in press) for informal species descriptions, and discussion
about which species are considered reworked and *in situ*.





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





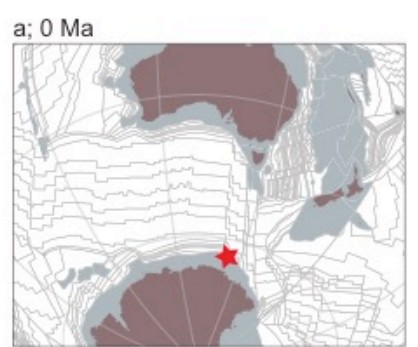

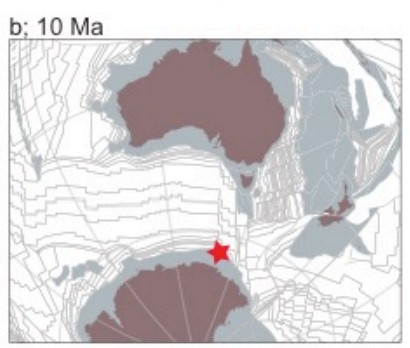

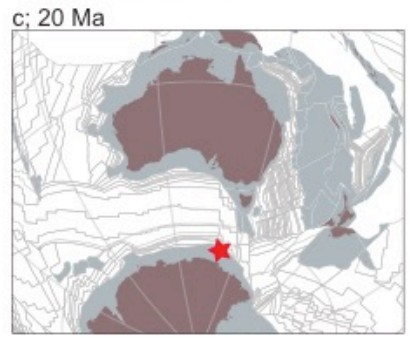

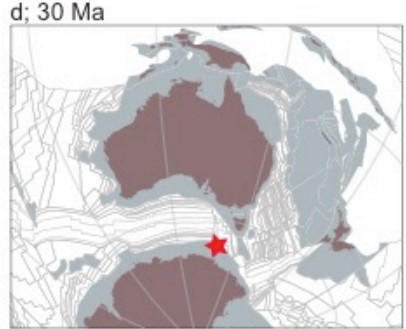



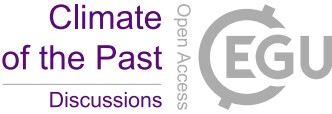

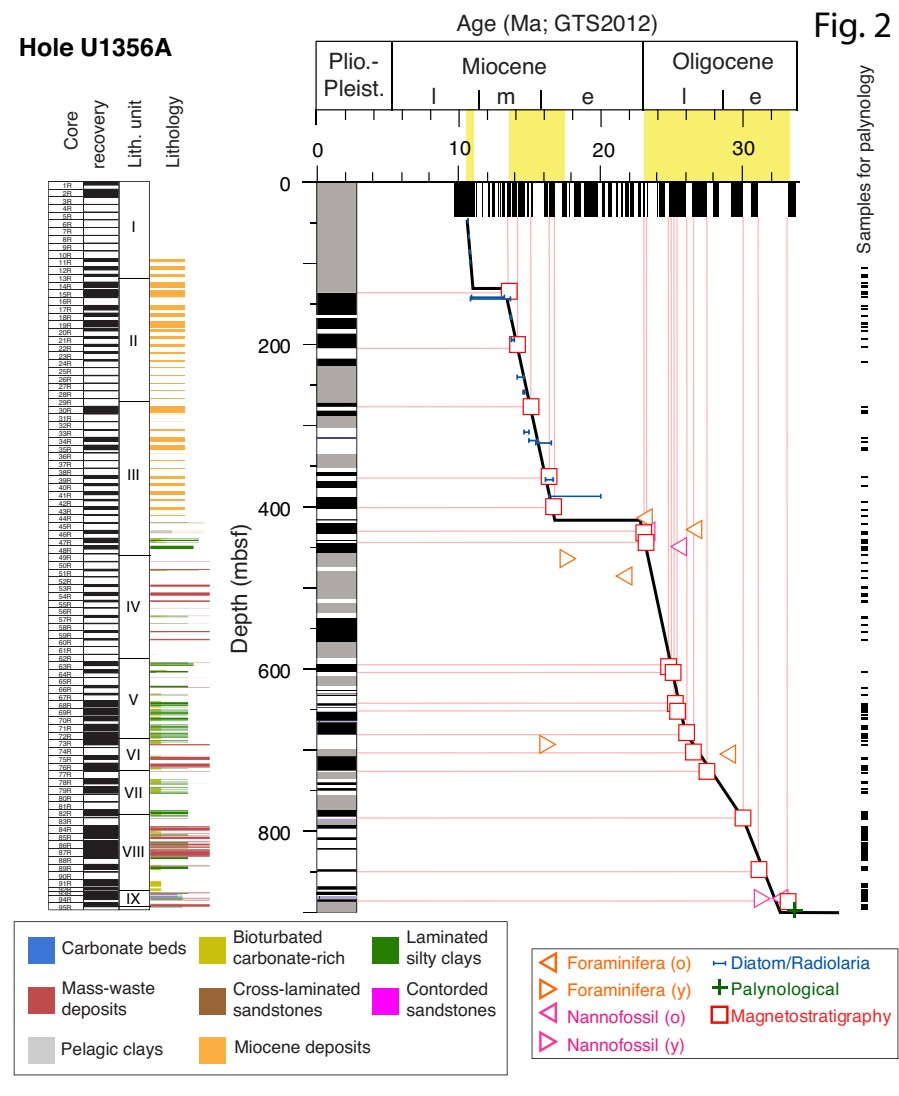







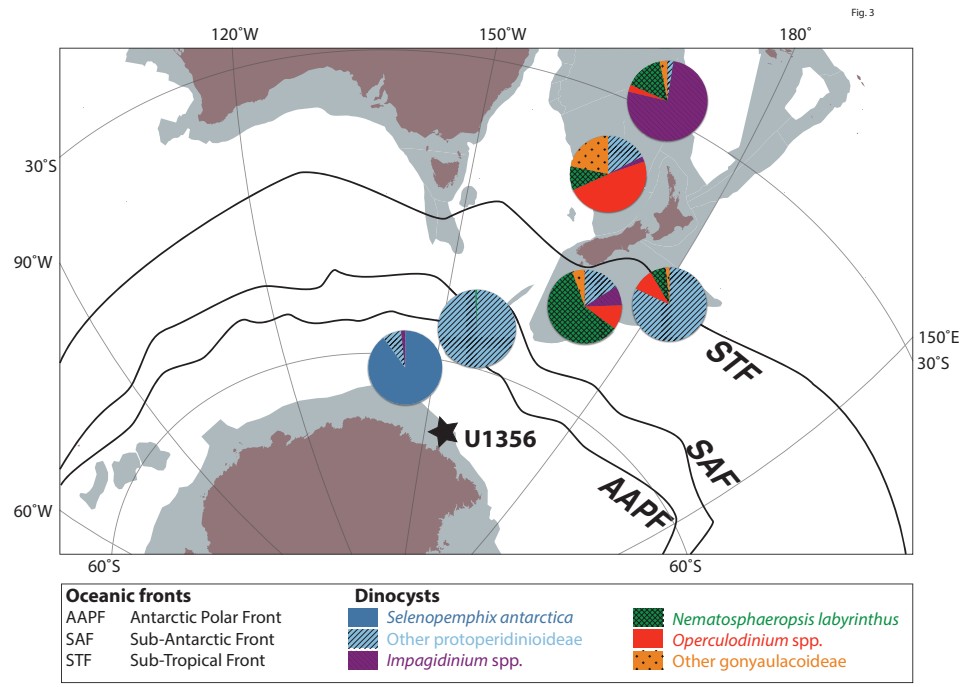











Fig. 5



Fig. 6

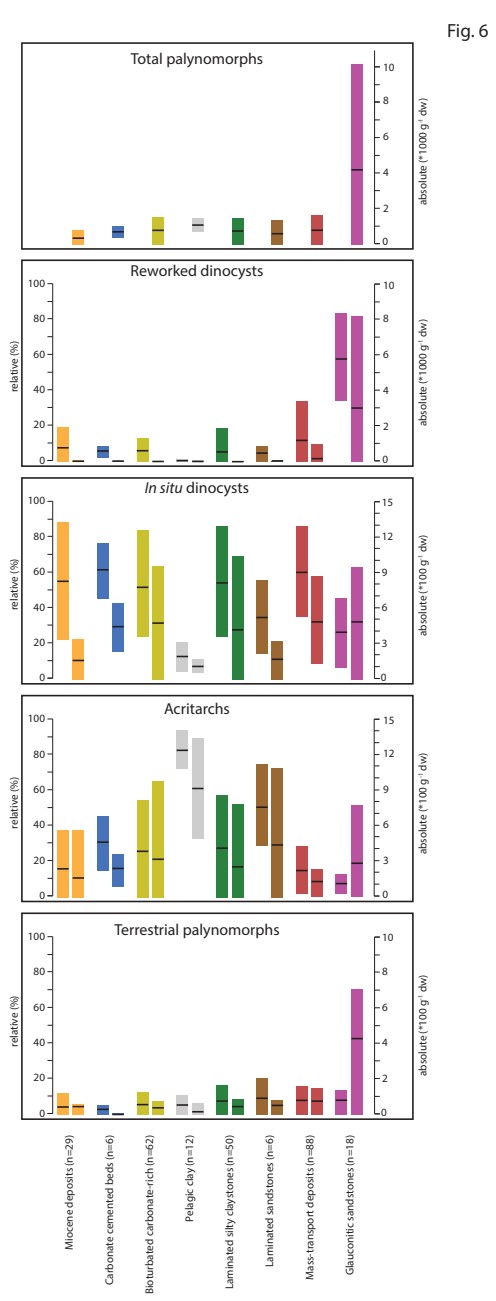



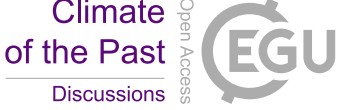



Fig. 7

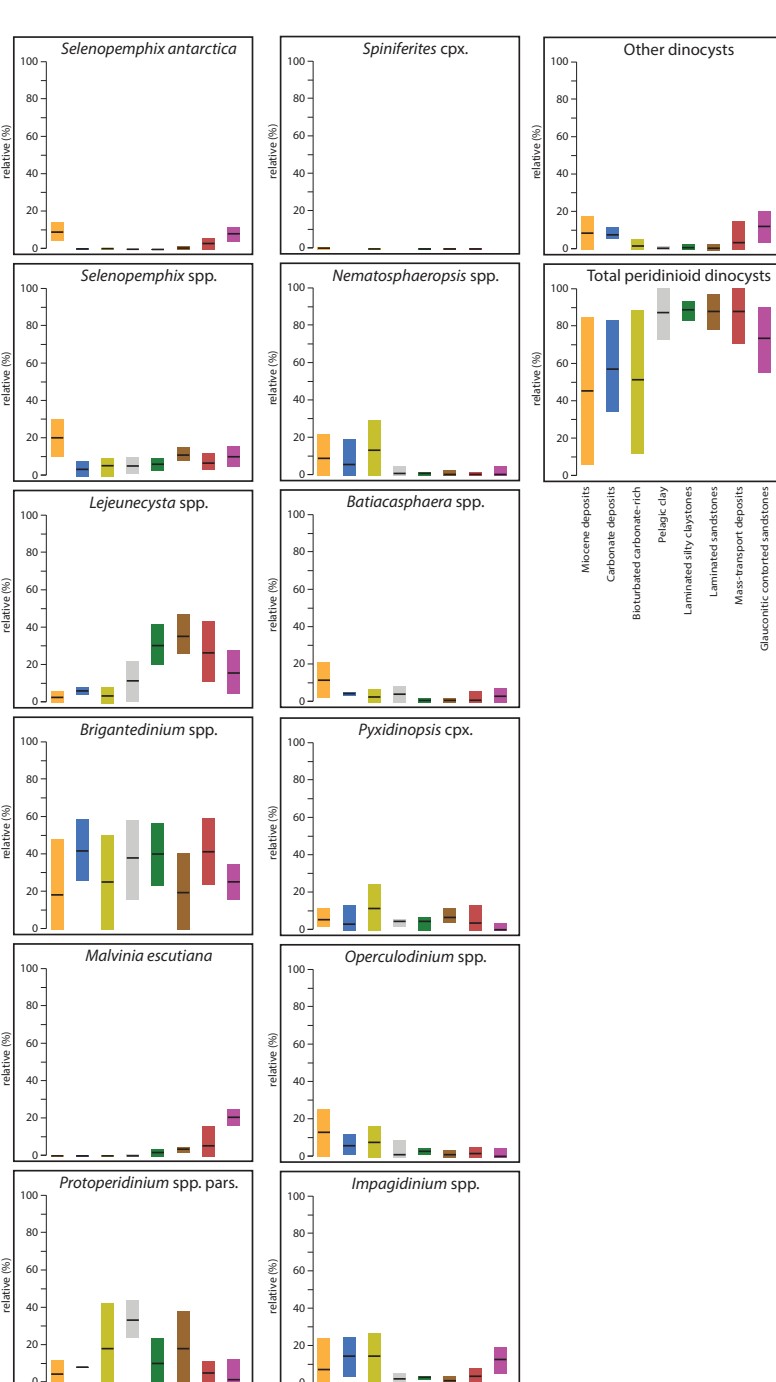



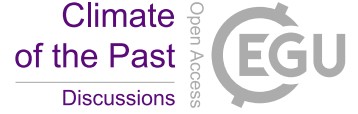

| type | FO/LO | Genus, chron | (Gradstein 20 | top core | top interval | bottom core | bottom inter | depth averag | error | Table 1 |
|------|-------|--------------|---------------|----------|--------------|-------------|--------------|--------------|-------|---------|
| CONOP | | | 10.76 | | | | | 98.66 | | |
| CONOP | | | 10.92 | | | | | 133.80 | | |
| CONOP | | | 13.41 | | | | | 133.81 | | |
| PM | (o) | C5ACn | 14.07 | 22R-2, | 75 | 22R-2, | 90 | 203.23 | 0.07 | |
| PM | (y) | C5Bn.2n | 15.03 | 30R-2, | 50 | 30R-2, | 75 | 279.63 | 0.13 | |
| PM | (o) | C5Cn.1n | 16.27 | 39R-1, | 35 | 39R-1, | 65 | 364.10 | 0.15 | |
| PM | (o) | C5Cn.3n | 16.72 | 42R-2, | 59 | 43R-1, | 25 | 398.28 | 3.98 | |
| | | | 17.50 | 44R-CC | | 45R-CC | | 416.90 | | |
| | | | 23.00 | 44R-CC | | 45R-CC | | 416.91 | | |
| PM | (o) | C6Cn.2n | 23.03 | 45R-CC | 40 | 46R-1 | 65 | 426.78 | 5.00 | |
| PM | (o) | C6Cn.3n | 23.30 | 50R-1, | 0 | | | 469.00 | 9.00 | |
| PM | (y) | C7An | 24.76 | 63R-3, | 85 | 63R-3, | 120 | 597.12 | 0.17 | |
| PM | (o) | C7An | 24.98 | 64R-1, | 130 | 64R-1, | 135 | 604.33 | 0.02 | |
| PM | (o) | C8n.1n | 25.26 | 68R-2, | 20 | 68R-2, | 75 | 643.38 | 0.27 | |
| PM | (y) | C8n.2n | 25.30 | 69R-2, | 20 | 69R-2, | 25 | 652.58 | 0.02 | |
| PM | (o) | C8n.2n | 25.99 | 71R-6, | 115 | 72R-1, | 10 | 678.98 | 0.92 | |
| PM | (y) | C9n | 26.42 | 73R-4, | 90 | 75R-1, | 15 | 701.66 | 7.09 | |
| PM | (o) | C9n | 27.44 | 76R-6, | 35 | 76R-6, | 40 | 725.09 | 0.02 | |
| PM | (o) | C11n.2n | 29.97 | 82R-6, | 35 | 82R-6, | 40 | 782.68 | 0.03 | |
| PM | (y) | C13n | 33.16 | 93R-1,117 | | 93R-2, | 28 | 878.00 | 0.23 | |


| in Salabarnada et al. (submitted this volume) | in this paper | Table 2 |
|-----------------------------------------------|---------------|---------|
| Laminated facies "F1" | Laminated siltstones | |
| | Laminated sandstones | |
| Bioturbated facies "F2" | Bioturbated carbonate-rich Pelagic clays | |
| Carbonate cemented beds | Carbonate cemented beds | |
| Turbidites and hemipelagites | Miocene deposits | |
| Slumps facies | | |
| Debris flows facies | Mass-waste deposits | |
| EOT facies | | |
| Eocene sands facies | Glauconite sandstones | |







| In situ taxa | Reworked taxa | Table 3 |
|---|---|---|
| Adnatosphaeridium? sp. | Achilleodinium biformoides | |
| Ataxodinium choane | Achomosphaera alcicornu | |
| Batiacasphaera compta | Aiora fenestrata | |
| Batiacasphaera spp. (pars.) | Aireiana verrucosa | |
| Batiacasphaera hirsuta | Adnatosphaeridium spp. | |
| Batiacasphaera micropapillata | Alisocysta circumtabulata | |
| Batiacasphaera minuta | Alterbidinium distinctum | |
| Batiacasphaera sphaerica | Apectodinium spp. | |
| Batiacasphaera sp. A | Arachnodinium antarcticum | |
| Batiacasphaera sp. B | Areoligera spp. (pars) | |
| Batiacasphaera sp. C | Areoligera semicirculata | |
| Batiacasphaera sp. D | Cerebrocysta bartonensis | |
| Brigantedinium simplex | Charlesdowniea clathrata | |
| Brigantedinium pynei | Charlesdowniea edwardsii | |
| Brigantedinium sp. A | Cooksonidinium capricornum | |
| Brigantedinium sp. B | Cordosphaeridium fibrospinosum | |
| Brigantedinium sp. C | Cordosphaeridium funiculatum | |
| Brigantedinium sp. D | Corrudinium incompositum | |
| Cerebrocysta WR small | Corrudinium regulare | |
| Cerebrocysta delicata | Cribroperidinium spp. | |
| Cerebrocysta sp. A | Damassadinium crassimuratum | |
| Cleistosphaeridium sp. B | Dapsilidinium spp. | |
| Cleistosphaeridium sp.A | Deflandrea sp. A sensu Brinkhuis et al., 2003 | |
| Cordosphaeridium minutum | Deflandrea antarctica | |
| Corrudinium labradori | Deflandrea cygniformis | |
| Corrudinium sp. A | Diphyes colligerum | |
| Cryodinium? sp. | Deflandrea spp. Indet | |
| Distatodinium spp. | Eisenackia circumtabulata | |
| Edwardsiella sexispinosa | Enneadocysta diktyostila | |
| Elytrocysta sp. A | Enneadocysta multicornuta | |
| Elytrocysta brevis | Eocladopyxis tesselata | |
| Gelatia inflata | Fibrocysta axialis | |
| Habibacysta? spp. | Glaphyrocysta intricta | |
| Homotryblium spp. | Glaphyrocysta pastielsii | |
| Hystrichokolpoma bullatum | Heteraulacacysta leptalea | |
| Huystrichosphaeropsis obscura | Histiocysta palla | |
| Impagidinium spp. (pars) | Hystrichokolpoma pusilla | |
| Impagidinium aculeatum | Hystrichokolpoma rigaudiae | |
| Impagidinium cantabrigiense | Hystrichokolpoma truncatum | |
| Impagidinium elegans | Hystrichosphaeridium truswelliae | |
| Impagidinium elongatum | Hystrichosphaeridium tubiferum | |
| Impagidinium pacificum | Impagidinium maculatum | |
| Impagidinium pallidum | Impagidinium waipawense | |
| Impagidinium paradoxum | Kenleyia spp. | |
| Impagidinium patulum | Manumiella druggii | |
| Impagidinium plicatum | Melitasphaeridium pseudorecurvatum | |
| Impagidinium velorum | Membranophoridium perforatum | |
| Impagidinium victorianum | Octodinium askiniae | |
| Impagidinium sp. A | Odontochitina spp. | |
| Impagidinium sphaericum | Operculodinium spp. | |
| Invertocysta tabulata | Phthanoperidinium antarcticum | |
| Islandinium spp. | Phthanoperidinium echinatum | |
| Lejeunecysta attenuata | Polysphaeridium spp. | |
| Lejeunecysta adeliense | Rhombodinium sp. | |
| Lejeunecysta fallax | Schematophora speciosa | |
| Lejeunecysta cowei | Schematophora obscura | |
| Lejeunecysta acuminata | Senegalinium spp. | |
| Lejeunecysta rotunda | Spinidinium luciae | |
| Lejeunecysta katatonos | Spinidinium macmurdoense | |
| Malvinia escutiana | Spinidinium schellenbergii | |
| Nematosphaeropsis labyrinthus | Spiniferites ramosus CPX | |
| Oligokolpoma galeotti | Thalassiphora pelagica | |
| Operculodinium tiara | Turbiosphaera filosa | |
| Operculodinium sp. A | Turbiosphaera sagena | |
| Operculodinium piaseckii | Vozzhennikovia apertura/ S.schellenbergii group | |
| Operculodinium janduchenei | Vozzhennikovia netrona | |
| Operculodinium cf eirikianum | ?Vozzhennikovia LARGE | |
| Operculodinium eirikianum | Wetzeliella articulata | |
| Paleocystodinium golzowense | | |
| Paucisphaeridium spp. | | |
| Phthanoperidinium amoenum | | |
| Pyxidinopsis spp. (pars) | | |
| Pyxidinopsis sp. A | | |
| Pyxidinopsis sp. B | | |
| Pyxidinopsis sp. C | | |
| Pyxidinopsis sp. D | | |
| Pyxidinopsis vesciculata | | |
| Pyxidinopsis tuberculata | | |
| Pyxidinopsis reticulata | | |
| Pyxidinopsis fairhavensis | | |
| Reticulatosphaera actinocoronata | | |
| Selenopemphix antarctica | | |
| Selenopemphix nephroides | | |
| Selenopemphix dioneacysta | | |
| Selenopemphix sp. A | | |
| Selenopemphix undulata | | |
| Selenopemphix brinkhuisi | | |
| Spiniferites sp. B | | |
| Spiniferites sp. A | | |
| Spiniferites sp. C | | |
| Stoveracysta ornata | | |
| Stoveracysta kakanuiensis | | |
| ?Svalbardella spp. | | |
| Tectatodinium spp. | | |
| Unipontedinium aquaeductus | | |
| Protoperidinioid indet | | |
| Protoperidinium sp. B | | |
| Protoperidinium sp. A | | |
| Protoperidinium sp. C | | |
| Protoperidinium sp. D | | |