# Peer review of "Oligocene–Miocene paleoceanography off the Wilkes Land Margin (East Antarctica) based on organic-walled dinoflagellate cysts"

_Climate of the Past, 2017_

## Referee Comment (RC1) · K. K. Sliwinska (Referee) · 29 Jan 2018

GENERAL COMMENTS At the beginning, I would like to apologize for the delay in delivering my review. It was great to get an opportunity to comment on this paper. For some time now, I have been working on the Oligocene from the North Atlantic region. Even though our study areas are so far from each other, one cannot fully understand the paleoclimatic changes in the high northern latitudes and the global ocean circulation under the early icehouse world, without an insight into the oceanic regime in the southern high latitudes. This paper provides an important and unique record of the paleogeographical reconstruction of the Oligocene to middle Miocene of the East Antarctica based on dinoflagellate cysts. Authors apply selected dinocysts genera and taxons as proxies for sea-ice reconstruction, nutrients, and temperature. The changes in the composition of dinocysts assemblages is additionally correlated with the sedimentology and organic biomarker data. I find this manuscript interesting and very needed piece of work for our understanding of the oceanic circulation under the early icehouse world conditions. A concern however, is the way the sedimentological data are incorporated into the text. The results of the present study (i.e. changes in the dinocysts assemblages) need to be clearly presented, and other data (sedimentology, biomarkers) should be carefully included but only as a data supporting the results based on dinocysts. The part about the lithology should not be included in the section with the results but as e.g. the background information. Also, a term "Miocene deposits" (Table 2) doesn't not carry any sedimentological information. Why do the authors not keep the terminology by Salabarnada et al. (submitted this volume) in this case? This expression is not used in the main text, but "Miocene sediments". The manuscript is well written, however, there is still room for improvement (see my suggestions below). Overall, the manuscript represents a substantial contribution to the scientific progress within the scope of Climate of the Past. I am certain that it will be of great interest for readers of the journal.

SPECIFIC COMMENTS In the Supplementary material, in the sheet with the dinocysts counts I see only Selenopemphix cf. antarctica. Is that a typo or the specimen observed in the present study only partially resemble the holotype? If it different, then I think that this needs a bit of attention in the text. Bijl et al. (in press) have already discussed which dinocysts are in situ and which not, so I think that the first section of the discussion can be tightened up a bit. Also, since dinocysts play a key role in this study, I would consider to include a plate with photos of the most important taxa.

Terrestrial palynomorphs can include everything from saccate-pollen to spores or fungal hyphae, and thus suggests e.g. a different depositional setting for the site. Therefore, I think that it may be a bit risky to put them into one category without mentioning

any details. One way to fix this is to give appropriate overheads in the "dinocysts counts" spreadsheet in the supplementary excel file (i.e. in situ dinocysts, reworked dinocysts, terrestrial palynomorphs, etc.) and refer to this file in the main text.

The strong upwelling occurring today around Antarctica is causing low abundances of carbonates at the sea-floor. How does the upwelling (suggested in line 363) support the presence of carbonate rich intervals during the Oligocene and Miocene (e.g. line 401)? I think that this needs to be explained a bit more clearly.

TECHNICAL CORRECTIONS Within the entire text "Margin" with a capital letter in "the Wilkes Land Margin". Please correct where needed. It needs to be clearly stated when the authors talk about "dinoflagellates" and when about "dinoflagellate cysts (dinocysts)". "sea-ice" or "sea ice", please choose only one version Please define: "common" or "abundant"

Abstract: Please avoid repetitions: "time intervals" line 25,27,44 Lines 25-29: "may bear information to resolve"? please rephrase the two sentences. Lines 37-38: Consider rephrasing to "Our record shows that a sea-ice indicator, Selenopemphix antarctica, occurs only in the earliest Oligocene, following the full Antarctic continental glaciation, and after the Middle Miocene Climatic Optimum". Line 39: "during the remainder of the . . ." – please rephrase Line 39: perhaps it is better to write: "the composition of the dinocyst assemblages imply"

Section 1: Line 51: please rephrase: ". . .much more ice is. . ." Lines 72-84: perhaps these two very long sentences could be made into few shorter ones. Lines 95-96: marine-ice? I think that "sea-ice" sounds better Line 96: does it mean "a continent with a low topography"? If yes, then please rephrase "a lower Antarctic" Line 115: please rephrase ". . .establishment of age control. . ." Line 125: perhaps "recently" instead of "accurately" Line 127-128: this sentence is poorly constructed Line 133-134: it sounds a bit weird to compare with "detailed sedimentological descriptions", I think that it should rather be written that the authors "correlate changes in the dinocyst assemblages with the changes in the lithology" or something like that. Line 135-139: this sentence is missing something. Please rephrase.

Section 2: Keep this section in the passive voice. Line 149: "upper Miocene" not "late Miocene" Line 165-170: this sentence is poorly constructed. It is not correct to write that "the lithology lacks" something Line 166-170: diatom ooze and diatom-rich clay: which one is a turbidite and or hemipelagite (see Table 2)? Line 178-179: this sentence is poorly constructed

Section 3: Line 196-197. Avoid active voice. Please rephrase both sentences. For me it sounds a bit weird to say "surface sample". What about "a sample from the sea surface" instead? "Another important information" is used in line 227 and 231. Consider rephrasing to avoid repetition. Line 235-236: What does "N" mean? I think it is better to write "north". Please rephrase the sentence to make it more clear. Please explain all the abbreviations used in the text for the first time, e.g. GCM, STF and SAF.

Section 4.1: Please describe the individual groups in the same order as they are mentioned at the beginning of the paragraph. Line 249-250: "amorphous organic matter (particles)" instead of "amorphous palynofacies". Line 252: it should be "rare to common" not "present to common". In this section it should also be explain how authors define: "rare, "common" and "abundant". Line 257: one can not write "dominate the assemblage during the late Oligocene". It should be either "are the dominating group in the assemblages from the upper Oligocene" or "were dominating/most abundant during the late Oligocene".

Section 4.2: Line 266: if it is not an observation made by the authors, I would suggest to add a reference here. Line 267-269: I suggest to rephrase the sentence: "is common to abundant between 33.6 to 32.1 Ma (earliest Oligocene) and after 14.2 Ma (i.e. during and after the mid-Miocene climatic transition)" Line 270: please remove "generally". Line 270-281: please consider to rephrase this part, so it will be clear what was the assemblage composition in the Oligocene-Miocene and what is today. Line 289: please

remove "noted" Line 291: Instead of "Of these taxa" it should be " Of the gonyaulacoid taxa" and add "spp." after Nematosphaeropsis.

Line 294: it should be Section 4.3 not 4.5. Please correct in the following headings accordingly, i.e. 4.3.1 and 4.3.2. Lines 296-306: I am not certain if the part describing the lithology fits in the result section. This is not a result of the current study, but rather a summary of the (already interpreted) lithological observations by Salabarnada et al. However, I see that this is an important part for the manuscript, I suggest to keep it, but incorporate it into the earlier part of the manuscript.

Section 4.5.1: Line 314: perhaps it should be: "...occur in the reworked glauconitic sandstones of the lower Oligocene age."? Line 315: Keep sentences short: "...sandstones. This is in line..." Line 316: Great, that what one can expect!

Section 4.5.2 Please, avoid expressions as "we compare", "we note", etc. Please change it into the passive voice. Lines 327-328: repetition of "interval" Line 330: "restricted to" or "limited to" instead of "connected to" Line 333: "in the Eocene sediments" Line 334-336: I suggest to rewrite like this: "Within the Oligocene strata Lejeunecysta spp. (...) lower abundance in the interglacial deposits and pelagic clays. The taxon is also less abundant in the Miocene."

Section 5. Discussion Line 353: why upwelling? Is that the only possibility? Lines 354-356: circular argumentation, that abundant oligotrophic cyst taxa support oligotrophic dinoflagellate assemblage Line 357: which taxa? It may be a good idea to list them here as a reminder for readers Line 359-362: "we interpret that these taxa are part of the in situ pelagic assemblage and reflect warming of surface waters rather than them being reworked" – I think that this needs rephrasing. What is more, which taxa are considered as indicators of warming? Is this based on the present study or the literature? If on the literature, then please provide proper references here. Line 366-367: this sentence is poorly constructed Lines 368-369: active voice should be avoided here Lines 370-372: grammatically something is missing in this sentence. Line 381: what does

"the average assemblage" means? Lines 387, 391: add "Site" before U1356 Line 391: please add "succession at Site U1356". Lines 393-394: repetition of lines 381-382 Line 365-396: it sounds weird to compare "Oligocene-Miocene surface waters" with "the same Oligocene-Miocene sediments". Please consider rewriting Line 407: "i.e." instead of "e.g." Line 420: "more oligotrophic character of the dinocyst assemblages" – please rephrase Line 430: "an evidence" Lines 449-450: this sentence is poorly constructed Line 451: modern dinocysts assemblages? Line 455: "...ACC. This is in line with numerical..." Line 460: please explain what does abbreviation MMCO means, perhaps even earlier in the text Line 465: consider different order, like: "weaker throughout the Oligocene and the Miocene, than at present" Line 467: please remove "to us" Line 476: please explain what does abbreviation MMCT means, perhaps earlier in the text Line 533: "records have recorded"- please rephrase

Section 6 Avoid repeating "fundamentally different" so close to each other (Lines 534 and 542), or "that of today" (line 542 and 543), "compared to today" (lines 548, 550 Lines 545-547: please consider rephrasing this sentence.

Line 608: it should be "data compiled from Site" Line 611: please use passive voice Line 613: perhaps it should be "or calibrating our data against age-scale" Line 622: "sandstones" – please correct in the entire text

Figure captions and references:

"Bijl et al. in press" not in the reference list "Salabarnada et al. submitted this volume" not in the reference list

Fig. 2 – Why does the colour lines reflecting various lithology have different length? What does (o) and (y) mean? Please align overheads "Miocene" and "Oligocene". Please explain what the grey colour in the palmag column implies. "(from Tauxe et al.,2012, but recalibrated to GTS2012 of Gradstein et al., 2012; see Table 1 and modified based on Crampton et al., 2016)" - this sentence is poorly constructed

Fig. 4 and 5 – what is determining the order of the dinocysts? Shouldn't Spiniferites cpx be moved to the right? And actually, is Spiniferites cpx needed on the figure if it is not even mentioned in the main text? The same with Corrudinium, Cerebrocysta – these are not mentioned in the text. If they are merged in a complex with Pyxidinopsis spp. then please clearly state it in the text or supplementary.

Fig. 4 – I think that it is necessary to mark the position of unconformities in e.g. the column with "epoch and stage". Otherwise, Chattian followed immediately by Burdigalian looks a bit odd. The intervals which look like barren in the column with " Total palynomorphs/dinocysts", are not marked as such in the following plots in the figure, therefore the figure looks a bit chaotic. The overheads for "total palynomorphs/dinocysts" and "Palynomorph relative abundance" should be aligned with the overheads to the right (i.e. dinocysts taxa and genera). Also, I would suggest to add a column with sample position on this and the following figure. Are all other dinocysts recorded in the assemblages "oligotrophic/outside oceanic fronts" as suggested by the color/filling in the plot? It is not clear to me why "oligotrophic/outside oceanic fronts" has two colors (red and dotted orange). Why are absolute abundances not shown in the same way as the relative abundances?

Fig. 5 – While in Fig.2 Oligocene and Miocene are divided into "late", "middle" and "early" , on figs 4 and 5 they are divided into stages. Adding a subdivision of the Oligocene and Miocene into "late", "middle" and "early" on figures 4 and 5 will help readers to directly correlate it with figure 2. This may be a good place to mark a position of the climatic events mentioned in the main text, such as the Oi-1 glaciation and MMCO. Please add that the figure shows the distribution of the "in situ dinocyst", like in figure 4.

Fig. 6-7: According to table 2 "Miocene deposits" consist partially of turbidites. Isn't that a bit odd that turbidite deposits yield so many in situ dinocysts?

Fig.7 – I would write something like that: "The distribution of eco-groups within various

lithologies encountered in Site. . ." in the figure caption.

With my best regards

Kasia K. Sliwinska

Please also note the supplement to this comment:
https://www.clim-past-discuss.net/cp-2017-148/cp-2017-148-RC1-supplement.pdf

---

## Referee Comment (RC2) · Anonymous Referee #2 · 1 Mar 2018

Oligocene-Miocene paloceanography off the Wilkes Land margin (East Antarctica) based on organic-walled dinoflagellate cysts.

Bijl et al.

Climate Past

Review 19 Feb. 2018

This paper presents environmental interpretations of a new dinoflagellate cyst dataset from Oligocene and Miocene sediments from a drill core collected off the Wilkes Land coast. The environmental interpretations are partly underpinned by published studies

on the distribution of dinoflagellate cysts in modern sea floor sediments. In particular, assemblages are identified that are interpreted to correlate with sea ice. The authors use these assemblages to conclude that sea ice was more prevelant during the earliest Miocene, and also following the Middle Miocene Climatic Optimum. They also observe that assemblages representative of interglacial conditions are similar to assemblages of modern temperate oligotrophic waters, and thus infer that this reflects a migration of the polar frontal system to the south of the drill site.

This is an interesting paper, and the dataset is important. It will be of interest to the research community.

I have four main comments on the approach used and the conclusions drawn

(1) The authors note that in modern settings, Selenopemphix antarctica is dominant in 'proximal sea ice settings south of the Antarctic Polar Front' (but also that these modern samples from Antarctic waters have a range of 10-90% S. antarctica). The authors then infer that the intervals in the Wilkes Land core containing the highest relative abundance of S. antarctica represent depositional environments proximal to sea ice. However, S. antarctica is never above ∼15% in any of the samples reported in this study (Figure 7): this taxon is not dominant. For context, samples with concentrations of up to 20% S. antarctica occur in modern Southern Ocean samples as far north as the Subtropical Front (e.g. Zonneveld et al. 2013, doi.org/10.1016/j.revpalbo.2012.08.003). Even if high abundance (>80%) of S. antarctica were indicative of sea ice (which is itself not clearly demonstrated, partly given the poor modern correlation between the polar front and sea ice extent, and partly due to the very sparse coverage of modern samples south of the polar front), that high abundance is not the case in the samples reported in this paper. The modern analogue approach used by the authors to infer the presence of sea ice is inconclusive in this instance: the data presented could be just as easily used to infer a complete lack of sea ice for the duration of the record, as sea ice variability.

(2) The authors conclude they demonstrate 'variability on glacial/interglacial timescales'. This is possibly true, but it has not been illustrated in a convincing way. The key to their interpretation, I think, is figures 6 and 7, where the relative abundance of different dinoflagellate cysts are illustrated for different lithologies. However, there is no evidence presented in this paper that these lithologies are deposited under different glacial conditions. They instead refer to Salabarnada et al. (in review submitted to CPD). Salabarnada et al. describe a glacial 'Facies 1', and an interglacial 'Facies 2'. Although the present authors rely on the cyclo-stratigraphy of Salabarnada et al. for their glacial-interglacial interpretation, they choose (confusingly) to apply a different lithological scheme in the present paper. Thus, in Table 2, the authors assign 'Silty claystones and sandstones' to (glacial) Facies 1 of Salabarnada et al., and 'carbonated rich and pelagic clay lithologies to (interglacial) Facies 2. Notwithstanding this, the dinoflagellate cyst assemblages shown in Figures 6 and 7 do not vary in a consistent way between either the glacial and interglacial facies described by Salabarnada et al., or by the glacial and interglacial lithologies assigned by the authors (line 300-302). The different lithologies do contain different dinoflagellate cyst assemblages, but these differences do not appear to fall along the glacial/interglacial divisions proposed by either Salabarnada et al. or the authors. However the authors choose to respond to this comment, at a minimum the abstract should be adjusted to removed the implication that glacial/interglacial has been investigated for the entire record (line 46), as only Oligocene samples have been explored for this variability, and I strongly suggest marking clearly on Figures 6 and 7 which lithologies represent glacial and which interglacial deposition, or perhaps grouping samples together - the seven columns/lithologies do not communicate clearly the variability the authors claim to have identified.

(3) The authors rely on unpublished (submitted, in review) work to justify their division of the dinoflagellate cyst assemblage into in situ and reworked components. This is an important step in their data processing, and important to completely assess this paper, but the information is not available to review at present.

(4) The discussion is fairly speculative/not well supported by the data presented – but is thought provoking, and should be retained.

Minor comments follow:

L299 relation not relations

L353 can the authors discount input of terrestrial nutrients instead of upwelling?

L422 replace 'a close position' with 'proximal'?

---

## Referee Comment (RC3) · S. Gallagher (Referee) · 14 Mar 2018

This is an excellent description and interpretation of organic microfossils from the Oligo-Miocene strata off Wilkes Land Margin and their palaeoceanographic significance.

I have made extensive comments and suggestions in the attached annotated pdf text to this paper.

I would like to add the following to the discussion:

I appreciate the utility of using isotopes to interpret Antarctic Ice Sheet variability as summarise by Liebrand et al 2017 (www.pnas.org/cgi/doi/10.1073/pnas.1615440114)

and this approach is used extensively when discussing the Cenozoic greenhouse ice-house transition. However, there are other sections that have been interpreted using backstripping and stratigraphic data in the Gippsland (Oligocene) and New Jersey (Oligo-Miocene) margins that reflect glacio-eustasy in the Oligocene and relative ice volume (Gallagher et al., 2013), it would be useful to consider the significance of these near field and far field sections in any section reviewing ice volume variability.

Gallagher, S. J., G. Villa, R. N. Drysdale, B. S. Wade, H. Scher, Q. Li, M. W. Wallace, and G. R. Holdgate (2013), A near-field sea level record of East Antarctic Ice Sheet instability from 32 to 27 Myr, Paleoceanography, 28, doi: 10.1029/2012PA002326.

More specific comments are below:

Lines 116-117 please clarify these lines they are bit vague.

Line 132: I don't think the word is speculate here, perhaps hypothesize?

Section 3 Methods: Bijl et al in press, I presume this means the paper published in Jl Micropal so please correct as it is not even in the reference list.

Line 217: what does empirical data mean here?

Paragraph starting at line 332: I found this section quite confusing, I have attempted to edit this in the annotated text.

5. Discussion section needs a few lines of a preamble.

Page 22 line 507 onward, as mentioned above there are other records that point to ice sheet instability in the time period being considered, please include in discourse.

Lines 518-527 are too speculative and should be left out or moderated.

In conclusion, once the text has been clarified and the suggestions considered this will be useful addition to the relatively sparsely documented Antarctic (palaeo)climate and oceanographic records.

Stephen Gallagher

Please also note the supplement to this comment:
https://www.clim-past-discuss.net/cp-2017-148/cp-2017-148-RC3-supplement.pdf

**Supplement:**

[Figure]

(subpolar) oceanic frontal systems have varied in concordance with Oligocene-

Miocene glacial-interglacial climate variability.

**1. Introduction**

[revised manuscript text omitted]

*also suggest*

submitted; Hartman et al., submitted this volume)  the same region

*a*                                                                *warmer*

*than today*

as modern analogue for the TEX$_{86}$ index values found (Hartman et al., submitted this

*therefore temperatures of*

volume) as for the dinocysts (Prebble et al., 2013);

*8 – 17 °C prevailed during*

 the Oligocene-Miocene at U1356.  thus

 warmer paleoceanographic regime close to

*Addition*

 Supporting evidence for

*warmer*                                          *is suggested by*

temperate Oligocene-Miocene surface waters comes from the abundance of

*in the strata*

nannofossils  Oligocene-Miocene  (Escutia et al.,

*from Nica     are rare*

2011b). Today, carbonate-producing plankton is  in high-latitude

*shells*

[revised manuscript text omitted]

*markedly*                    *The*

weaker during Oligocene and Miocene

*when*        *1    5   were smaller*

today.  continental ice sheet

The strength of the influence of

*follows*          *water spreads*

warm oligotrophic surface water  deep-sea $\delta^{18}O$ values: With enhanced low-latitude influence of surface water during times of light $\delta^{18}O$ in the deep sea and *vice versa*. The absence of (a trend towards more) oceanographic isolation of the Wilkes Land margin throughout the Oligocene to mid-Miocene

*may have attained its*

suggests that the ACC  not obtain  full, present-day strength until  the mid-Miocene Climatic transition.

with stronger influence of oligotrophic, low-

*prevailed              alternating with*

latitude surface waters over Site U1356 during interglacial times  more eutrophic,

*related to the*

colder influence during glacial times.  latitudinal migration of the AAPF

**Acknowledgements**

This research used data and samples from the Integrated Ocean Drilling Program (IODP). IODP was sponsored by the U.S. National Science Foundation and participating countries under management of Joined Oceanographic Institutions Inc.

PKB and FS thank NWO-NNPP grant no 866.10.110, NWO-ALW VENI grant no

863.13.002 for funding and Natasja Welters for technical support. CE and AS thank the Spanish Ministerio de Economía y Competitividad for Grant CTM2014-60451-C2-

1-P.

---

## Author Comment (AC1) · 10 Apr 2018

GENERAL COMMENTS At the beginning, I would like to apologize for the delay in delivering my review. It was great to get an opportunity to comment on this paper. For some time now, I have been working on the Oligocene from the North Atlantic region. Even though our study areas are so far from each other, one cannot fully understand the paleoclimatic changes in the high northern latitudes and the global ocean circulation under the early icehouse world, without an insight into the oceanic regime in the southern high latitudes. This paper provides an important and unique record of the paleogeographical reconstruction of the Oligocene to middle Miocene of the East Antarctica based on dinoflagellate cysts. Authors apply selected dinocysts genera and taxons as proxies for sea-ice reconstruction, nutrients, and temperature. The changes in the composition of dinocysts assemblages is additionally correlated with the sedimentology and organic biomarker data. I find this manuscript interesting and very needed piece of work for our understanding of the oceanic circulation under the early icehouse world conditions. A concern however, is the way the sedimentological data are incorporated into the text. The results of the present study (i.e. changes in the dinocysts assemblages) need to be clearly presented, and other data (sedimentology, biomarkers) should be carefully included but only as a data supporting the results based on dinocysts. The part about the lithology should not be included in the section with the results but as e.g. the background information. Also, a term "Miocene deposits" (Table 2) doesn't not carry any sedimentological information. Why do the authors not keep the terminology by Salabarnada et al. (submitted this volume) in this case? This expression is not used in the main text, but "Miocene sediments". The manuscript is well written, however, there is still room for improvement (see my suggestions below). Overall, the manuscript represents a substantial contribution to the scientific progress within the scope of Climate of the Past. I am certain that it will be of great interest for readers of the journal.

**We appreciate the positive assessments by Śliwińska regarding our manuscript, and her indications as to how to improve our manuscript even further. Śliwińska posed several concerns and suggestions, which we can definitely use to improve our manuscript. We herein respond to these concerns and suggestions in detail.**

SPECIFIC COMMENTS In the Supplementary material, in the sheet with the dinocysts counts I see only Selenopemphix cf. antarctica. Is that a typo or the specimen observed in the present study only partially resemble the holotype? If it different, then I think that this needs a bit of attention in the text.
**This is indeed a typo, it does fall within the species definition of the holotype. We will amend this in our next version of the paper.**

Bijl et al. (in press) have already discussed which dinocysts are in situ and which not, so I think that the first section of the discussion can be tightened up a bit.
**The first section of our discussion aims at providing the necessary details to put forward new arguments than those proposed in Bijl et al., in press (now Bijl et al., 2018) to strengthen and support the reason why we believe that the gonyaulacoid dinocysts are *in situ*. Therefore we do not find this redundant but rather complimentary to the results of Bijl et al., 2018, as indicated in lines 366-368. This paper targets a different audience than that of Journal of Micropaleontology, an audience that does not necessarily want to read detailed micropaleontological contemplations, but is merely interested in the paleoceanographic reconstructions. Such reconstructions are based on detailed micropaleontological information that is now published in Bijl et al., 2018, should the reader be interested. Journal of Micropalaeontology is an open access journal, hence available to everyone. Because of the above, we opt for maintaining the first section of the manuscript as**

**is.**

Also, since dinocysts play a key role in this study, I would consider to include a plate with photos of the most important taxa.
**Bijl et al. (2018, Journal of Micropalaeontology) also features a large number of dinocyst plates, and the publication is open access. This paper however is targeted to present the paleoceanographic reconstructions, using the dinocysts as a tool rather than the purpose of the study. With that aim in mind, and anticipating on the audience expected, we decided that plates are irrelevant in this paper. However, we added reference to the plates as published in Bijl et al. 2018 in the methods section (3.1)**

Terrestrial palynomorphs can include everything from saccate-pollen to spores or fungal hyphae, and thus suggests e.g. a different depositional setting for the site. Therefore, I think that it may be a bit risky to put them into one category without mentioning any details. One way to fix this is to give appropriate overheads in the "dinocysts counts" spreadsheet in the supplementary excel file (i.e. in situ dinocysts, reworked dinocysts, terrestrial palynomorphs, etc.) and refer to this file in the main text.
**An extensive presentation of the terrestrial palynology and the vegetation and climate reconstructions derived from it, is out of the scope of this paper, and will be presented elsewhere at a later stage. For the purposes of our paper, we portray the total terrestrial organic component in our samples as a crude and qualitative proxy for terrestrial input. Since details of terrestrial palynomorphs are meant to be presented in another study, we only recorded broad categories of terrestrial palynomorphs in our counts, which we present in the figure and in the supplementary tables.**

The strong upwelling occurring today around Antarctica is causing low abundances of carbonates at the sea-floor. How does the upwelling (suggested in line 363) support the presence of carbonate rich intervals during the Oligocene and Miocene (e.g. line 401)? I think that this needs to be explained a bit more clearly.
**This is explained around lines 429-433, where the oceanographic reconstructions are discussed.**

TECHNICAL CORRECTIONS Within the entire text "Margin" with a capital letter in "the Wilkes Land Margin". Please correct where needed.
**We will change 'Margin' to lower case throughout**

It needs to be clearly stated when the authors talk about "dinoflagellates" and when about "dinoflagellate cysts (dinocysts)". "sea-ice" or "sea ice", please choose only one version
**We will check throughout for consistency**

Please define: "common" or "abundant"
**We will rephrase throughout and specify to avoid ambiguity.**

Abstract: Please avoid repetitions: "time intervals" line 25,27,44 **Done** Lines 25-29: "may bear information to resolve"? **Rephrased** please rephrase the two sentences. Lines 37-38: Consider rephrasing to "Our record shows that a sea-ice indicator, Selenopemphix antarctica, occurs only in the earliest Oligocene, following the full Antarctic continental glaciation, and after the Middle Miocene Climatic Optimum". **Done** Line 39: "during the remainder of the : : :" – please rephrase Line 39: perhaps it is better to write: "the composition of the dinocyst assemblages imply" **Rephrased**

Section 1: Line 51: please rephrase: ": : :much more ice is: : :" **Rephrased** Lines 72-84:

perhaps these two very long sentences could be made into few shorter ones. **Sentences were shortened** Lines 95-96: marine-ice? I think that "sea-ice" sounds better **We talk about marine-based ice and not sea ice in those lines, which have a rather different meaning.** Line 96: does it mean "a continent with a low topography"? If yes, then please rephrase "a lower Antarctic" **Done** Line 115: please rephrase ": : :establishment of age control: : :" **Rephrased** Line 125: perhaps "recently" instead of "accurately" **Rephrased** Line 127-128: this sentence is poorly constructed **Rephrased** Line 133-134: it sounds a bit weird to compare with "detailed sedimentological descriptions", I think that it should rather be written that the authors "correlate changes in the dinocyst as- semblages with the changes in the lithology" or something like that. **Rephrased** Line 135-139: this sentence is missing something. Please rephrase. **Rephrased**

Section 2: Keep this section in the passive voice. **We used passive voice more than in the previous manuscript, but not in every case to avoid a too passive tone, which to our opinion does not read well.** Line 149: "upper Miocene" not "late Miocene" **Rephrased** Line 165-170: this sentence is poorly constructed. It is not correct to write that "the lithology lacks" something **Rephrased** Line 166-170: diatom ooze and diatom-rich clay: which one is a turbidite and or hemipelagite (see Table 2)? **We agree that our initial analyses lacked a detailed description of the Miocene facies. In the new version of the manuscript we will add the detailed Miocene lithology to the Oligocene one. We have already made this amendment in anticipation of this rebubuttal and noticed, however, that this does not affect our conclusions and drawn earlier.** Line 178-179: this sentence is poorly constructed **Rephrased**

Section 3: Line 196-197. Avoid active voice. **Avoided in most cases.** Please rephrase both sentences. For me it sounds a bit weird to say "surface sample". What about "a sample from the sea surface" instead? **We agree with the comment and will rephrase surface-samples to surface-sediment samples.** "Another important information" is used in line 227 and 231. Consider rephrasing to avoid repetition. **Rephrased** Line 235-236: What does "N" mean? I think it is better to write "north". **Done** Please rephrase the sentence to make it more clear. Please explain all the abbreviations used in the text for the first time, e.g. GCM, STF and SAF. **Checked and done**

Section 4.1: Please describe the individual groups in the same order as they are mentioned at the beginning of the paragraph. **We will change the order.** Line 249-250: "amorphous organic matter (particles)" instead of "amorphous palynofacies". **Done** Line 252: it should be "rare to common" not "present to common". **Rephrased** In this section it should also be explain how authors define: "rare, "common" and "abundant". **Rephased to avoid ambiguity** Line 257: one can not write "dominate the assemblage during the late Oligocene". It should be either "are the dominating group in the assemblages from the upper Oligocene" or "were dominating/most abundant during the late Oligocene". **Rephrased**
Section 4.2: Line 266: if it is not an observation made by the authors, I would suggest to add a reference here. **Done** Line 267-269: I suggest to rephrase the sentence: "is common to abundant between 33.6 to 32.1 Ma (earliest Oligocene) and after 14.2 Ma (i.e. during and after the mid-Miocene climatic transition)" **Done** Line 270: please remove "generally". **Done** Line 270-281: please consider to rephrase this part, so it will be clear what was the assemblage composition in the Oligocene-Miocene and what is today. **Rephrased** Line 289: please remove "noted" **Done** Line 291: Instead of "Of these taxa" it should be " Of the gonyaulacoid taxa" and add "spp." after Nematosphaeropsis. **Changed to *N. labyrinthus.*** Line 294: it should be Section 4.3 not 4.5. Please correct in the following headings accordingly, i.e. 4.3.1 and 4.3.2. **Done** Lines 296-306: I am not certain if the part describing the lithology fits in the result section. This is not a result of the current study, but rather a summary of the (already interpreted) lithological observations by

Salabarnada et al. However, I see that this is an important part for the manuscript, I suggest to keep it, but incorporate it into the earlier part of the manuscript. **Indeed, lithological details can be avoided and we now refer to Salabarnada et al., for details.** Section 4.5.1: Line 314: perhaps it should be: ": : :occur in the reworked glauconitic sandstones of the lower Oligocene age."? **Done** Line 315: Keep sentences short: ": : :sandstones. This is in line: : :" **Done** Line 316: Great, that what one can expect! Section 4.5.2 Please, avoid expressions as "we compare", "we note", etc. Please change it into the passive voice. **Done** Lines 327-328: repetition of "interval" **Rephrased** Line 330: "restricted to" or "limited to" instead of "connected to" **Rephrased** Line 333: "in the Eocene sediments" **done**
Line 334-336: I suggest to rewrite like this: "Within the Oligocene strata Lejeunecysta spp. (: : :) lower abundance in the interglacial deposits and pelagic clays. The taxon is also less abundant in the Miocene." **Rephrased**

Section 5. Discussion Line 353: why upwelling? Is that the only possibility? **We believe that, given the geographic setting, upwelling is the only possibility. We now indicate that more clearly in the text** Lines 354- 356: circular argumentation, that abundant oligotrophic cyst taxa support oligotrophic dinoflagellate assemblage **Rephrased to avoid circular argumentation** Line 357: which taxa? It may be a good idea to list them here as a reminder for readers **We really want the reader to focus on the paleoceanographic inferences. As we have elaborately described the species in the results section, we do not repeat the species names here.** Line 359-362: "we interpret that these taxa are part of the in situ pelagic assemblage and reflect warming of surface waters rather than them being reworked" – I think that this needs rephrasing. **Done** What is more, which taxa are considered as indicators of warming? Is this based on the present study or the literature? If on the literature, then please provide proper references here. **Done** Line 366-367: this sentence is poorly constructed **Rephrased** Lines 368-369: active voice should be avoided here **Avoided** Lines 370-372: grammatically something is missing in this sentence. **Rephrased** Line 381: what does "the average assemblage" means? **Rephrased** Lines 387, 391: add "Site" before U1356 **Done** Line 391: please add "succession at Site U1356". **Done** Lines 393-394: repetition of lines 381-382 **Repetition avoided** Line 365-396: it sounds weird to compare "Oligocene-Miocene surface waters" with "the same Oligocene-Miocene sediments". Please consider rewriting **Agreed. Rephrased** Line 407: "i.e." instead of "e.g." **Done** Line 420: "more oligotrophic character of the dinocyst assemblages" – please rephrase **Rephrased** Line 430: "an evidence" **Done** Lines 449-450: this sentence is poorly constructed **Rephrased** Line 451: modern dinocysts assemblages? **Rephrased** Line 455: ": : :ACC. This is in line with numerical: : :" **Done** Line 460: please explain what does abbreviation MMCO means, perhaps even earlier in the text **Spelled out** Line 465: consider different order, like: "weaker throughout the Oligocene and the Miocene, than at present" **Done** Line 467: please remove "to us" **Done** Line 476: please explain what does abbreviation MMCT means, perhaps earlier in the text **Done** Line 533: "records have recorded"- please rephrase **Done**

Section 6 Avoid repeating "fundamentally different" so close to each other (Lines 534 and 542), or "that of today" (line 542 and 543), "compared to today" (lines 548, 550) **Done** Lines 545-547: please consider rephrasing this sentence. **Done** Line 608: it should be "data compiled from Site" **Rephrased** Line 611: please use passive voice **Done** Line 613: perhaps it should be "or calibrating our data against age-scale" **Rephrased** Line 622: "sandstones" – please correct in the entire text **Done**

Figure captions and references:
"Bijl et al. in press" not in the reference list "Salabarnada et al. submitted this volume" not in the reference list. **We added these references**

Fig. 2 – Why does the colour lines reflecting various lithology have different length? **This was done to improve clarity** What does (o) and (y) mean? **Now explained in the caption** Please align overheads "Miocene" and "Oligocene". **Done** Please explain what the grey colour in the palmag column implies. **Now explained in the caption**"(from Tauxe et al.,2012, but recalibrated to GTS2012 of Gradstein et al., 2012; see Table 1 and modified based on Crampton et al., 2016)" - this sentence is poorly constructed **Rephrased**

Fig. 4 and 5 – what is determining the order of the dinocysts? Shouldn't Spiniferites cpx be moved to the right? **Agreed, done** And actually, is Spiniferites cpx needed on the figure if it is not even mentioned in the main text? **Yes it is, as it is one of the most common dinocyst genera in many places.** The same with Corrudinium, Cerebrocysta – these are not mentioned in the text. If they are merged in a complex with Pyxidinopsis spp. then please clearly state it in the text or supplementary. **Now mentioned in the text.**

Fig. 4 – I think that it is necessary to mark the position of unconformities in e.g. the column with "epoch and stage". Otherwise, Chattian followed immediately by Burdigalian looks a bit odd. **Done** The intervals which look like barren in the column with " Total palynomorphs/ dinocysts", are not marked as such in the following plots in the figure, therefore the figure looks a bit chaotic. **Many barren samples are positioned close to productive samples. The plot is meant to provide the reader with a comprehensive image of the palynological assemblages, similarly to the way they were presented and discussed in the text.** The overheads for "total palynomorphs/dinocysts" and "Palynomorph relative abundance" should be aligned with the overheads to the right (i.e. dinocysts taxa and genera). **Done** Also, I would suggest to add a column with sample position on this and the following figure. **The sample intervals are already plotted in Figure 2. We believe that this information is no longer needed when interpreting the data in figures 4 and 5.** Are all other dinocysts recorded in the assemblages "oligotrophic/outside oceanic fronts" as suggested by the color/filling in the plot? **We clarified this in the results section in the text.** It is not clear to me why "oligotrophic/outside oceanic fronts" has two colors (red and dotted orange). **We choose to give *Operculodinium* spp. another color because it is such a well-known and paleoceanographically significant genus both in this region and in the northern hemisphere.** Why are absolute abundances not shown in the same way as the relative abundances? **Absolute abundances of the different dinocyst groups are not mentioned or discussed in the text, nor they do have a readily interpretable paleoceanographic signal.**

Fig. 5 – While in Fig.2 Oligocene and Miocene are divided into "late", "middle" and "early" , on figs 4 and 5 they are divided into stages. Adding a subdivision of the Oligocene and Miocene into "late", "middle" and "early" on figures 4 and 5 will help readers to directly correlate it with figure 2. **Agreed. Done** This may be a good place to mark a position of the climatic events mentioned in the main text, such as the Oi-1 glaciation and MMCO. **Agreed. Done** Please add that the figure shows the distribution of the "in situ dinocyst", like in figure 4. **Done in the caption**

Fig. 6-7: According to table 2 "Miocene deposits" consist partially of turbidites. Isn't that a bit odd that turbidite deposits yield so many in situ dinocysts? **We agree and thought about this. Possibly, turbidites in the Miocene transport very young sediments from the shelf. This causes reworking in these turbidites to be overlooked as there is no age gap between the species encountered in the turbidites from those encountered in the pelagic sediments. We will add this to the main text of the paper, and in any case we now separate turbidite deposits from pelagic sediments.** However, Fig.7 – I would write something like that: "The distribution of eco-groups within various lithologies encountered in Site: : :" in the figure caption. **Done**

With my best regards

Kasia K. Sliwinska

Please also note the supplement to this comment:
https://www.clim-past-discuss.net/cp-2017-148/cp-2017-148-RC1-supplement.pdf

---

## Author Comment (AC2) · 10 Apr 2018

This paper presents environmental interpretations of a new dinoflagellate cyst dataset from Oligocene and Miocene sediments from a drill core collected off the Wilkes Land coast. The environmental interpretations are partly underpinned by published studies on the distribution of dinoflagellate cysts in modern sea floor sediments. In particular, assemblages are identified that are interpreted to correlate with sea ice. The authors use these assemblages to conclude that sea ice was more prevalent during the earliest Miocene [**We assume R2 means Oligocene here**], and also following the Middle Miocene Climatic Optimum. They also observe that assemblages representative of interglacial conditions are similar to assemblages of modern temperate oligotrophic waters, and thus infer that this reflects a migration of the polar frontal system to the south of the drill site. This is an interesting paper, and the dataset is important. It will be of interest to the research community.

I have four main comments on the approach used and the conclusions drawn

(1) The authors note that in modern settings, Selenopemphix antarctica is dominant in 'proximal sea ice settings south of the Antarctic Polar Front' (but also that these modern samples from Antarctic waters have a range of 10-90% S. antarctica). The authors then infer that the intervals in the Wilkes Land core containing the highest relative abundance of S. antarctica represent depositional environments proximal to sea ice. However, S. antarctica is never above _15% in any of the samples reported in this study (Figure 7): this taxon is not dominant. **[Fig. 7 only reports the mean and 1sd of the data. The maximum abundance of *S. antarctica* is 39%. We will make the raw data available in a revised version]** For context, samples with concentrations of up to 20% S. antarctica occur in modern Southern Ocean samples as far north as the Subtropical Front (e.g. Zonneveld et al. 2013, doi.org/10.1016/j.revpalbo.2012.08.003). Even if high abundance (>80%) of S. antarctica were indicative of sea ice (which is itself not clearly demonstrated, partly given the poor modern correlation between the polar front and sea ice extent, and partly due to the very sparse coverage of modern samples south of the polar front), that high abundance is not the case in the samples reported in this paper. The modern analogue approach used by the authors to infer the presence of sea ice is inconclusive in this instance: the data presented could be just as easily used to infer a complete lack of sea ice for the duration of the record, as sea ice variability. **We agree with the reviewer that the complete compilation by Prebble et al. (2013) leaves ambiguity about the reliability of *S. antarctica* as sea ice indicator, and that the absence of this species should be taken as absence of sea. Sites south of the subtropical front with lower abundances of *S. antarctica* are all close to the polar front itself, and are in regions with lower palaeobathymetry (e.g., Kerguelen and in the South Atlantic). This causes highly variable distribution patterns around such bathymetric highs (see, e.g., Armand et al., 2008). Meanwhile, on the Antarctic continental shelf proper, where admittedly few published data is available in the Prebble et al. (2013) compilation, *Selenopemphix antarctica* does dominate the palynomorph assemblages in all sites available. The dominance of *S. antarctica* in assemblages can be found in the Wilkes Land margin itself (Site U1357; Hartman et al., in prep-a), in the Ross Sea (Hartman et al., in prep-b), Prydz Bay (Storkey, 2006), in the Indian Ocean (Marret and De Vernal, 1997) and in the Weddell Sea (Esper and Zonneveld, 2002; Harland and Pudsey, 1999). We echo the studies from Houben et al. (2013) and Sangiorgi et al. (2018), which elaborately discus the potential of *S. antarctica* as sea-ice indicator and its ecological meaning. We understand that the explanation in our manuscript falls short in providing the reader sufficient information on this matter. In a new version of the manuscript, we will support our inference of *S. antarctica* as sea ice indicator (and its absence as indicator of longer-than-today open water season) more elaborately than we did so far.**

(2) The authors conclude they demonstrate 'variability on glacial/interglacial timescales'. This is possibly true, but it has not been illustrated in a convincing way. The key to their interpretation, I think, is figures 6 and 7, where the relative abundance of different dinoflagellate cysts are illustrated for different lithologies. However, there is no evidence presented in this paper that these lithologies are deposited under different glacial conditions. They instead refer to Salabarnada et al.

(in review submitted to CPD). Salabarnada et al. describe a glacial 'Facies 1', and an interglacial 'Facies 2'. Although the present authors rely on the cyclo-stratigraphy of Salabarnada et al. for their glacial-interglacial interpretation, they choose (confusingly) to apply a different lithological scheme in the present paper. Thus, in Table 2, the authors assign 'Silty claystones and sandstones' to (glacial) Facies 1 of Salabarnada et al., and 'carbonated rich and pelagic clay lithologies to (interglacial) Facies 2. Notwithstanding this, the dinoflagellate cyst assemblages shown in Figures 6 and 7 do not vary in a consistent way between either the glacial and interglacial facies described by Salabarnada et al., or by the glacial and interglacial lithologies assigned by the authors (line 300-302). The different lithologies do contain different dinoflagellate cyst assemblages, but these differences do not appear to fall along the glacial/interglacial divisions proposed by either Salabarnada et al. or the authors.

**We agree that the different presentation of the lithologic facies in our ms and that of Salabarnada et al may generate confusion. In a new version of the manuscript, we will make this consistent. In anticipation of this review, we have already revisited the Miocene lithology, made a detailed description and integrated the facies into the other lithologies described in Salabarnada et al. This did not lead to any different conclusions than those already made, namely a higher relative abundance of protoperidinioid dinocysts in glacial deposits, and more gonyaulacoid dinocysts in interglacial deposits, with the lithologic interpretations being made independent of the dinocyst results in Salabarnada et al, CP (https://doi.org/10.5194/cp-2017-152).**

However [**if**] the authors choose to respond to this comment, at a minimum the abstract should be adjusted to removed the implication that glacial/interglacial has been investigated for the entire record (line 46), as only Oligocene samples have been explored for this variability, and I strongly suggest marking clearly on Figures 6 and 7 which lithologies represent glacial and which interglacial deposition, or perhaps grouping samples together - the seven columns/lithologies do not communicate clearly the variability the authors claim to have identified.

**We agree with the reviewer, a new version of the manuscript will present the dinocyst data in fewer lithologic groups. Moreover, the detailed lithologic interpretations will be continued into the Miocene part of the sequence. This will only reinforce the interpretations of different dinocyst assemblages between glacial and interglacial deposits.**

(3) The authors rely on unpublished (submitted, in review) work to justify their division of the dinoflagellate cyst assemblage into in situ and reworked components. This is an important step in their data processing, and important to completely assess this paper, but the information is not available to review at present.

**The paper is now published and available open access in Journal of Micropalaeontology.**

(4) The discussion is fairly speculative/not well supported by the data presented – but is thought provoking, and should be retained.

**Because the reviewer does not substantiate which part he/she finds speculative, we cannot reply any further to this comment at this stage. We will thoroughly revisit the discussion and evaluate any speculative aspects.**

Minor comments follow:
L299 relation not relations -**done**
L353 can the authors discount input of terrestrial nutrients instead of upwelling? **We can for most of the record, with reason and argument, not with unequivocal proof. Given the relatively small catchment area, and deteriorated climate, the low relative abundance of palynomorphs (those that are there are mostly wind-transported pollen) and absence of terrestrially-derived amorphous organic matter, and the average outer neritic/oceanic nature of the dinocyst assemblage, we argue for marine nutrients instead of terrestrially-derived. Although, the Miocene Climatic Optimum might have an additional terrestrially-derived nutrient source. We shall add this to the manuscript.**

L422 replace 'a close position' with 'proximal'? **This was not found, possible lost in revision References:**

Esper, O. and Zonneveld, K.A.F.: Distribution of organic-walled dinoflagellate cysts in surface sediments of the Southern Ocean (eastern Atlantic sector) between the Subtropical Front and the Weddell Gyre, Marine Micropaleotology, 46, 177-208, 2002.

Harland, R. and Pudsey, C. J., 1999. Dinoflagellate cysts from sediment traps deployed in the Bellingshausen, Weddell and Scotia seas, Antarctica. Marine Micropaleontology. 37, 77-99.

Hartman, J.D., Bijl, P.K., Sangiorgi, F., et al., in prep-a. Palynological assemblages from the Holocene of IODP Site U1357A, Wilkes Land margin, Antarctica. to be submitted to Journal of Micropaleontology

Hartman, J.D., Sangiorgi, F., Bijl et al., in prep-b. A multi-proxy reconstruction for MIS5 to MIS9 of the Antarctic marginal ice zone in the Ross Sea: sea-ice cover, productivity and temperature for Site AS05-10, Drygaski basin. to be submitted to Paleoceanography and Paleoclimatology.

Houben, A.J.P., Bijl, P.K., Pross, J., Bohaty, S.M., Passchier, S., Stickley, C.E., Röhl, U., Sugisaki, S., Tauxe, L., Van De Flierdt, T., Olney, M., Sangiorgi, F., Sluijs, A., Escutia, C., Brinkhuis, H.: Reorganization of Southern Ocean plankton ecosystem at the onset of Antarctic glaciation, Science, 340, 341-344, 2013.

Marret, F. and De Vernal, A., 1997. Dinoflagellate cyst distribution in surface sediments of the southern Indian Ocean. Marine Micropaleontology. 29, 367-392.

Prebble, J. G., Crouch, E. M., Carter, L., Cortese, G., Bostock, H., Neil, H., 2013. An expanded modern dinoflagellate cyst dataset for the Southwest Pacific and Southern Hemisphere with environmental associations. Marine Micropaleontology. 101, 33-48.

Sangiorgi, F., Bijl, P.K., Passchier, S., Salzmann, U., Schouten, S., McKay, R., Cody, R.D., Pross, J., Van De Flierdt, T., Bohaty, S.M., Levy, R., Williams, T., Escutia, C., Brinkhuis, H.: Southern Ocean warming and Wilkes Land ice sheet retreat during the mid-Miocene, Nature Communications, 9 (1), art. no. 317, 2018.

Storkey, C.A.: Distribution of marine palynomorphs in surface sediments, Prydz Bay, Antarctica. MSc thesis Victoria University of Wellington, New Zealand. http://hdl.handle.net/10063/21, 2006.

---

## Author Comment (AC3) · 10 Apr 2018

We appreciate the time and effort of Dr. S. Gallagher to review our manuscript. In fact, his comments significantly improved our manuscript. We agree a discussion about the comparison of our results with sea-level reconstructions from the Gippsland Basin would improve the overall picture of Southern OCean paleoceanographic conditions. Further textual suggestions will be incorporated in a next version of the manuscript.

Please also note the supplement to this comment:
https://www.clim-past-discuss.net/cp-2017-148/cp-2017-148-AC3-supplement.pdf

[Figure]

[Figure]

**Supplement:**

This is an excellent description and interpretation of organic microfossils from the Oligo-Miocene strata off Wilkes Land Margin and their palaeoceanographic significance. I have made extensive comments and suggestions in the attached annotated pdf tex to this paper.
I would like to add the following to the discussion:
I appreciate the utility of using isotopes to interpret Antarctic Ice Sheet variability as summarised by Liebrand et al 2017 (www.pnas.org/cgi/doi/10.1073/pnas.1615440114) and this approach is used extensively when discussing the Cenozoic greenhouse icehouse transition. However, there are other sections that have been interpreted using backstripping and stratigraphic data in the Gippsland (Oligocene) and New Jersey (Oligo-Miocene) margins that reflect glacio-eustasy in the Oligocene and relative ice volume (Gallagher et al., 2013), it would be useful to consider the significance of these near field and far field sections in any section reviewing ice volume variability. Gallagher, S. J., G. Villa, R. N. Drysdale, B. S. Wade, H. Scher, Q. Li, M. W. Wallace, and G. R. Holdgate (2013), A near-field sea level record of East Antarctic Ice Sheet instability from 32 to 27 Myr, Paleoceanography, 28, doi: 10.1029/2012PA002326.
**This is a good suggestion, we will add this to the revised manuscript.**

More specific comments are below:
Lines 116-117 please clarify these lines they are bit vague. **-done**
Line 132: I don't think the word is speculate here, perhaps hypothesize? **-done**
Section 3 Methods: Bijl et al in press, I presume this means the paper published in Jl Micropal so please correct as it is not even in the reference list. **indeed. Done**
Line 217: what does empirical data mean here? **Now better explained**
Paragraph starting at line 332: I found this section quite confusing, I have attempted to edit this in the annotated text.
5. Discussion section needs a few lines of a preamble. **Agreed. Done**
Page 22 line 507 onward, as mentioned above there are other records that point to ice sheet instability in the time period being considered, please include in discourse.
Lines 518-527 are too speculative and should be left out or moderated.
In conclusion, once the text has been clarified and the suggestions considered this will be useful addition to the relatively sparsely documented Antarctic (palaeo)climate and oceanographic records.

Please also note the supplement to this comment:
https://www.clim-past-discuss.net/cp-2017-148/cp-2017-148-RC3-supplement.pdf –**We will carefully incorporate the comments that were annotated into a new version of the ms.**

---

## Author Response (AR2)

**Universiteit Utrecht**

**Editorial Board, Climate of the Past**

To the editorial Board

**Faculty of Geosciences,**
**Department of Earth Sciences**
Marine Palynology and
Paleoceanography

**Visitors Address**
Vening Meineszgebouw A
Princetonlaan 8A
3584 CB Utrecht
The Netherlands

| | |
|---|---|
| **Your reference** | |
| **Our reference** | Manuscript to Climate of the Past |
| **Phone** | +31 30 253 9318 |
| **Fax** | +31 30 253 5096 |
| **Email** | p.k.bijl@uu.nl |
| **Website** | www.uu.nl/staff/pkbijl |
| **Date** | 26-06-2018 |
| **Subject** | Submission of manuscript to Climate of the Past |

Dear Professor Thornalley, Dear editor,

Please find enclosed our edited final revised manuscript, entitled "**Oligocene–Miocene paleoceanography off the Wilkes Land margin (East Antarctica) based on organic-walled dinoflagellate cysts**" (cp-2017-148).

We follow the review report, and have made the following final edits to the figures and the captions:
- Fig. 3: made the colour coding consistent with that of Fig. 4 and 5.
- Fig. 4. and 5: as suggested by the reviewer, we separated out Selenopemphix antarctica and Selenopemphix spp. pars
- Fig. 5: we adjusted the d18O compilation, which now shows the megasplice of De Vleeschouwer et al., 2017. We adjusted the caption accordingly.
- Fig. 6: as suggested by the reviewer, we split relative abundances from absolute abundances in two collumns
- Fig. 7: as suggested by the reviewer, we made a full-page figure, and we also separated out Selenopemphix antarctica from the other Selenopemphix species.

The final version of the manuscript still contains 7 figures, 2 tables and about 6500 words.

In anticipation of your reply,

Best regards, also on behalf of my co-authors,

Peter K. Bijl
Corresponding author